# Attitude-Tracking Control for Over-Actuated Tailless UAVs at Cruise Using Adaptive Dynamic Programming

**Zihou He \***[ID]**, Jianbo Hu, Yingyang Wang \*, Jiping Cong** [ID]**, Yuan Bian and Linxiao Han**

Equipment Management and Unmanned Aerial Vehicle Engineering College, Airforce Enginnering University, Xi'an 710072, China
\* Correspondence: hezihou9551@163.com (Z.H.); wangyingyang@outlook.com (Y.W.)

**Abstract:** Using adaptive dynamic programming (ADP), this paper presents a novel attitude-tracking scheme for over-actuated tailless unmanned aerial vehicles (UAVs) that integrates control and control allocation while accounting for nonlinearity and nonaffine control inputs. The proposed method uses the idea of nonlinear dynamic inversion to create an augmented system and converts the optimal tracking problem into an optimal regulation problem using a discounted performance function. Drawing inspiration from incremental control, this method achieves optimal tracking control for the nonaffine system by simply using a critic-only structure. Moreover, the unique design of the performance function ensures robustness against model uncertainties and external disturbances. The ADP method was found to outperform traditional control architectures that separate control and control allocation, achieving the same level of attitude-tracking performance through a more optimized approach. Furthermore, unlike many recent optimal controllers for nonaffine systems, our method does not require any model identifiers and demonstrates robustness. The superiority of the ADP-based approach is verified through two simulated scenarios, and its internal mechanism is further discussed. The theoretical analysis of robustness and stability is also provided.

**Keywords:** attitude tracking; tailless unmanned aerial vehicle; adaptive dynamic programming; UAV flight simulation; nonlinear flight control





## 1. Introduction

The tailless unmanned aerial vehicle (UAV) has garnered immense attention due to its promising potential in both civil and military aviation. Its superior aerodynamic efficiency in comparison to traditional designs offers benefits such as improved voyage, carrying capacity, and stealth performance. This has led to the emergence of several tailless UAVs such as Boeing's X-45 A, X-45B/C, Lockheed Martin's RQ-170 Sentinel, BAE's Taranis, and Dassault's NEURON.

The Innovative Control Effector (ICE) aircraft [1,2], developed through research by Lockheed Martin, stands out among the latest tailless aerial vehicles. ICE is equipped with as many as 11 effectors relocated compactly on its main wing, which makes the system over-actuated. Such a unique layout was designed to investigate and measure the aerodynamics and performance of various low-observable tailless configurations using innovative control effectors. After decades of research, ICE has been found to have excellent maneuverability and stealth performance, making it a good choice for future UAV design.

The unique configuration of ICE allows its effectors to achieve goals beyond providing aerodynamic moments, such as minimizing drag or maximizing lift [3]. However, this configuration also poses challenges for control. Like other tailless vehicles, ICE suffers from problems such as poor static stability and coupling between longitudinal and lateral dynamics. Additionally, the redundant effectors of ICE require dealing with the control allocation problem, which involves selecting appropriate effectors and providing deflection commands to generate the required moments. However, the compact layout of ICE's

effectors results in strong coupling effects between them, causing the control inputs to appear nonlinear in the system, which means that the system is nonaffine. As a result, the control allocation of ICE is an extremely challenging task.

Since the proposal of the concept of ICE, researchers in the flight control field have been paying constant attention to it [4–6]. In 2017, Niestroy et al. [7] published detailed aerodynamic data for ICE, which enabled the construction of a highly precise control-oriented nonlinear model and the development of advanced control algorithms. Many researchers have developed different control algorithms for the nonlinear ICE model. In a recent study, He et al. [8] proposed an altitude tracker for ICE using the well-known decoupling conditions for nonaffine systems [9–14], while other researchers prefer incremental control methods.

The principle behind incremental control methods is timescale separation, which makes use of Taylor expansion. Incremental control methods can transform nonaffine systems into incremental affine forms. Therefore, the complexity of nonlinear optimization in CA can be avoided. This approach is highly effective for dealing with nonaffine systems [15]. Recently, there have been several advancements in incremental control for ICE. Stolk et al. [3] proposed a minimum drag CA method based on incremental nonlinear dynamic inversion. Matamoros [16] implemented an incremental nonlinear CA in ICE, resulting in improved tracking and CA performance. Sun et al. [17] improved the CA of ICE using hierarchical multi-objective optimization and adaptive incremental backstepping. Additionally, He et al. [14] extended the incremental control to the outer-loop control of ICE trajectory tracking using the pseudo-control hedging technique and relaxing the need for a timescale separation principle.

The reason incremental control is effective for nonaffine systems is it makes good use of the partial derivative of control inputs, $\frac{\partial f(x,u)}{\partial u}$. This is obtained through the digital differentiation of aerodynamic data. It is worth noting that obtaining the partial derivative of control inputs may be challenging in some systems. However, with the advancements in wind-tunnel tests, we can obtain more accurate and economic aerodynamic data for flight control. When combined with model identification techniques, incremental control has great potential for the future.

Most of the aforementioned control methods separate the command tracking and control allocation (CA) tasks. The command tracker provides the aerodynamic coefficients command $\tau_c$ to ensure that the reference signal $x_c$ is accurately tracked, while the CA determines the specific effector deflection $u$ based on the aerodynamic coefficients command and its objective function. This framework is highly convenient for incorporating established flight control algorithms, and the CA can be viewed as an optimization problem that can leverage the well-established optimization theory. As a result, this framework is preferred by most researchers.

However, there are some aspects of the above framework that could be improved. The obvious drawback is that the CA is designed to minimize the objective function consisting of the moments tracking error $\tau_e = \tau - \tau_c$ and the effector deflection $u$. From an input and output perspective, the moments tracking error is only an intermediate value, and what truly matters is the reference signal tracking error $x_e = x - x_c$. Therefore, the ideal objective function should take both $x_e$ and the effector deflection into consideration, instead of just $\tau_e$. Additionally, most existing flight control algorithms are based on Lyapunov theory and can only take into account the convergence of $x_e$. To make these algorithms compatible with the over-actuated UAV, the above framework must be adopted, and the second goal is left to the CA.

Meanwhile, the above framework takes two steps to give effector deflection, increasing the computational time. Hou et al. [18] introduced the recurrent neural network in CA and claimed that the recurrent neural network model could be solved in parallel to meet the real-time requirement. Still, this approach has only been validated in a linearized model, where the computational load is inherently small.

Hence, it is imperative to develop more reasonable and effective frameworks that abandon the meaningless intermediate values, therefore ensuring convergence of $x_e$ and achieving the second goal in one step. Optimal control is a promising option in this regard. Unlike the objective function of CA that only considers $\tau_e$ and the second goal, the performance function of optimal control can incorporate the tracking error and any other desired second goals. This means that the command tracking and CA can be described using a single equation. However, solving the nonlinear Hamilton–Jacobi–Bellman (HJB) equation remains a formidable challenge.

Adaptive dynamic programming(ADP) provides new ideas for solving the nonlinear HJB equation. ADP is a heuristic algorithm for solving optimal control. Compared with other heuristic algorithms, such as reinforcement learning, ADP is supported by optimal control theory, so it shows better convergence and stability and is more suitable for flight control. The first application of ADP in optimal control could be seen in a study by Werbos [19]. ADP's basic idea is to use sampling data to drive a neural network to approximate the optimal value function. In this way, APD turns the backward-in-time dynamic programming process into a forward-in-time manner and greatly expands the application of optimal control. For theoretical studies of ADP, Wei and Liu [20] give the stability analysis of policy iterative APD, and the stability proof of value iterative ADP is given by Al-Tamimi and Lewis [21]. Moreover, the researchers also proposed different frameworks of ADP, such as heuristic dynamic programming [22], dual heuristic programming [23], and globalized dual heuristic programming [24]. These studies lay the foundation of ADP, and a more detailed review of recent studies on ADP can be found in the paper by Liu et al. [25]

Model identification is a commonly adopted technique in recent applications of ADP in practical systems [26,27]. Model identification is an effective method for enhancing the robustness of ADP, but it requires introducing an identifier network. Compared to basic ADP, which uses only a critical network to approximate the value function, the incorporation of additional networks significantly increases the computational complexity. Therefore, approaches to alleviate the computational burden, such as the event-trigger technique [28], are necessary for these methods.

However, it is often overlooked that the optimal control itself could be robust with an appropriate design of performance function. This idea is illustrated in a book by Lin [29], and the author systematically discusses how to handle disturbance and model uncertainty in an optimal control way. This way, the ADP could enjoy robustness while avoiding heavy computational burdens. However, the author also points out that it is still an open question how to apply similar approaches to a nonaffine system. Most ADPs are developed to address the optimal regulation problem, but for flight control, the optimal tracking control has a more practical use. With the development of the aviation industry, modern flight control is no longer satisfied with just ensuring flight stability. Many researchers [30–34] began to consider how to track the command signal optimally.

In the control field, optimal tracking control has attracted increased attention. One of the most common optimal tracking methods is the combination of feedforward control and feedbackward control [35–40]. The feedforward control is a traditional steady-state tracking controller to ensure the command reference signal is tracked. ADP is used in the feedbackward control to stabilize the transient error optimally. With the help of the traditional steady-state tracking controller, this optimal tracker shows good stability. Nevertheless, this optimal tracker is not suitable for ICE. Designing a traditional steady-state tracking controller for ICE has already been an arduous task, and the control allocation still needs to be considered in this process.

Some ADP-based optimal trackers do not rely on the feedforward control [41–43]. These studies applied a discounted performance function to ensure the boundness of the optimal value function in the infinite-time process and constructed an augmented system using the error dynamic and reference signal dynamic to transform the tracking problem into the regulation problem. However, the dynamic of the reference signal is unavailable in these methods, limiting the use of these ADPs.

From the other view, nonlinear dynamic inversion [44], as a tried-and-tested control method in the flight control area, constructs the dynamic of the desired signal using state value and command signal, which provides a new idea to overcome this drawback. Moreover, these ADPs also cannot address the nonaffine system, so they cannot be directly used for ICE.

To apply ADP in ICE, the nonaffine control input must be considered. Recent optimal trackers for nonaffine systems can be grouped into two types. One type is mainly for single-input systems, which decouple the nonaffine system into an affine system with model uncertainties [45–47], but it is not easy to extend such a method to the multi-input system. Using decoupling conditions, these methods show robustness, but neglecting model details also makes their optimization performance poor. The other type uses the other neural network, known as the actor network, to handle nonlinearity in control input and update the policy through gradient-base algorithm [48–50]. This method performs well but also needs more data and training, which undoubtedly increases the computational burden. Of course, some tricks commonly used in reinforcement learning [34] could also help improve the convergence rate of the method, but this also causes the lack of stability proofs.

Motivated by the aforementioned studies, this article proposes a critic-only ADP technique for the attitude tracking of ICE featured by nonaffine control inputs and redundant effectors. Through ADP, our approach integrates control and control allocation so that the same performance can be achieved at a cost less than conventional methods. By the idea of nonlinear dynamic inversion, an augmented system is constructed. The optimal tracking problem is transformed into an optimal regulation problem with discounted performance function, and the command dynamic is avoided. Inspired by the successful use of $\frac{\partial f(x,u)}{\partial u}$ in incremental control, we introduce $\frac{\partial f(x,u)}{\partial u}$ into APD, letting our method handle the nonaffine system in a simple way. Moreover, this article proves that for the control of the nonaffine system, the robust tracking problem could be equivalent to the optimal tracking problem with an augmented cost. This provides another way to improve the robustness of ADP, and complex model identification methods can be avoided.

The rest of the paper is arranged as follows: Section 2 introduces the aerodynamic model of the UAV. Section 3 gives the problem formulation and shows at the theoretical level that the optimal control with a specially designed performance function is equivalent to robust control. Section 4 presents the control scheme and stability analysis. Section 5 presents two simulations that validate the superiority of our method over the conventional approach and demonstrate its robustness, respectively. Finally, Section 6 gives the conclusion and the outlook for the next steps of research.

## 2. Model Description

This section introduces the ICE model. The basic parameters of ICE can be found in Table 1 [7], while more detailed information on the modeling of effectors can be found in Chapter 3 of [3]. Due to space constraints, this information will not be repeated here.

Figure 1 displays the layout of ICE, which features a high-sweep, tailless flying wing with a leading-edge sweep of 65 deg and 25 deg chevron shaping on the trailing edge. ICE is equipped with 13 independent effectors, including two pairs of leading-edge flaps (LEF), a pair of spoiler slot deflectors (SSD), a pair of all-moving tips (AMT), a pair of elevon (ELE), a pair of ganged pitch flaps (PF), and multi-axis thrust vectoring (MTV). Since this paper is focused on the cruising stage, MTV will not be taken into account.

The deflection ranges of the effectors are, inboard LEF: 0–40 deg, outboard LEF: ±40 deg, ELE: ±30 deg, PF: ±30 deg, AMT: ±60 deg, SSD: 0–60 deg. The rate limits on the leading-edge devices are 40 deg/s and on all the other surfaces 150 deg/s.

The modeling of the UAV is based on the following two assumptions: 1st, the UAV flies in the atmosphere, and the atmosphere is incompressible; 2nd, the UAV's body is rigid. Please note that only the body of the UAV is considered a rigid body, but the effectors are deformable.

**Table 1.** The basic parameters of ICE.

| Parameter | Nomenclature | Value | Unit |
|---|---|---|---|
| $b$ | Lateral–directional reference length, span | 37.50 | ft |
| $\bar{c}$ | Mean aerodynamic chord | 28.75 | ft |
| $m$ | Weight | 32,750 | LBF |
| $I_{yy}$ | Pitch Moment of Inertia | 78,451 | slug·ft$^2$ |
| $I_{xz}$ | Cross Product of Inertia | −525 | slug·ft$^2$ |
| $I_{xx}$ | Roll Moment of Inertia | 35,479 | slug·ft$^2$ |
| $I_{zz}$ | Yaw Moment of Inertia | 110,627 | slug·ft$^2$ |
| $S$ | Reference area | 808.60 | ft$^2$ |
| $X_{th}$ | Moment arm for thrust vectoring | 18.75 | ft |
| $X_{cg}$ | Gravity center | 38.84%$\bar{c}$ | |
| $X_{ac}$ | Aerodynamic center | 38.00%$\bar{c}$ | |

**Remark 1.** *In previous studies [8,44,51], the MTV was used solely during the UAV's vigorous maneuvers or when other effectors were saturated. However, this paper proposes a cruise-oriented approach where ADP enables a superior trade-off between effector deflection and tracking error. This effectively eliminates the need for MTV, ensuring that effector saturation is avoided.*

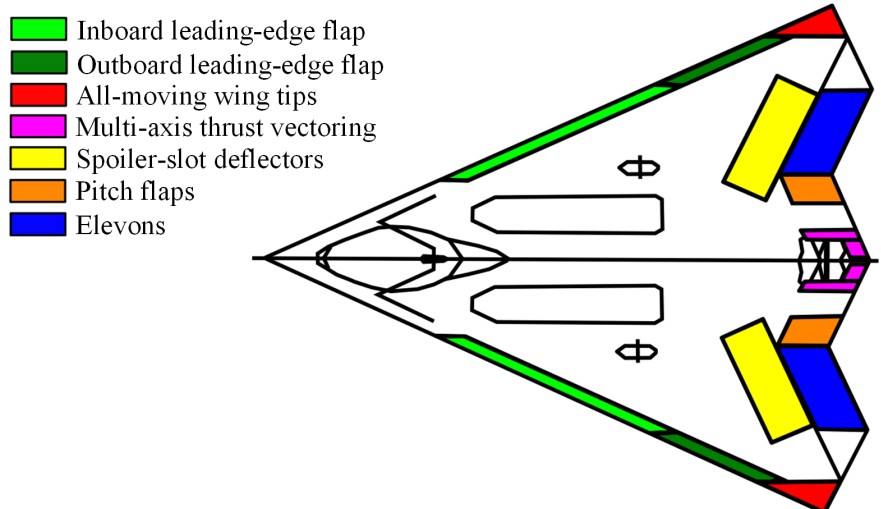

- 🟩 Inboard leading-edge flap
- 🟢 Outboard leading-edge flap
- 🟥 All-moving wing tips
- 🟪 Multi-axis thrust vectoring
- 🟨 Spoiler-slot deflectors
- 🟧 Pitch flaps
- 🟦 Elevons

**Figure 1.** The layout of ICE.

The motion equation of 6-DOF UAV model is given below[14,18,52], the nomenclature of the variables in the following equation can be found in Table 2.

$$\begin{bmatrix} \dot{V} \\ \dot{\chi} \\ \dot{\gamma} \end{bmatrix} = \begin{bmatrix} \frac{1}{M} & 0 & 0 \\ 0 & \frac{1}{MVc_\gamma} & 0 \\ 0 & 0 & \frac{-1}{MV} \end{bmatrix} \cdot \begin{bmatrix} T_{ve}G_f + F \end{bmatrix} \tag{1}$$

where $V$, $\gamma$, and $\chi$ are airspeed, flight path angle, and ground tracking angle, respectively. $s_\star$ and $c_\star$ represent $\sin \star$ and $\cos \star$. $F$ donate the sum of aerodynamic force and thrust, which could be approximated through accelerometers, and $G_f = [0\ 0\ Mg]^T$ represents gravitational forces. Define $\mu$, $\alpha$, and $\beta$ as the bank angle, angle of attack, and sideslip angle, then the dynamic of $[\mu\ \alpha\ \beta]$ is:

$$\begin{bmatrix} \dot{\mu} \\ \dot{\alpha} \\ \dot{\beta} \end{bmatrix} = \begin{bmatrix} c_\alpha c_\beta & 0 & s_\alpha \\ s_\beta & 1 & 0 \\ s_\alpha c_\beta & 0 & -c_\alpha \end{bmatrix}^{-1} \left( -T_{bv}^T \begin{bmatrix} -\dot{\chi}s_\gamma \\ \dot{\gamma} \\ \dot{\chi}c_\gamma \end{bmatrix} + \begin{bmatrix} p \\ q \\ r \end{bmatrix} \right) \tag{2}$$

where $p, q$, and $r$ are the body-axis roll, pitch, and yaw rates. The expression $T_{vb}$ is the transformation matrix from the body frame to the velocity frame, and $T_{ve}$ is the transformation matrix from the earth frame to the velocity frame. These matrices are given in [53]:

$$
T_{vb} = \begin{bmatrix} c_\alpha c_\beta & s_\beta & s_\alpha c_\beta \\ -c_\alpha s_\beta c_\mu + s_\alpha s_\mu & c_\beta c_\mu & -s_\alpha s_\beta c_\mu - c_\alpha s_\mu \\ -c_\alpha s_\beta s_\mu - s_\alpha c_\mu & c_\beta s_\mu & -s_\alpha s_\beta s_\mu + c_\alpha c_\mu \end{bmatrix}
\tag{3}
$$

$$
T_{ve} = \begin{bmatrix} c_\chi c_\gamma & s_\chi c_\gamma & s_\gamma \\ -s_\chi & c_\chi & 0 \\ c_\chi s_\gamma & s_\chi s_\gamma & c_\gamma \end{bmatrix}
\tag{4}
$$

The dynamic of $[p\ q\ r]$ is:

$$
\begin{bmatrix} \dot{p} \\ \dot{q} \\ \dot{r} \end{bmatrix} = J^{-1} \left( M_a - \begin{bmatrix} p \\ q \\ r \end{bmatrix} \times J \begin{bmatrix} p \\ q \\ r \end{bmatrix} \right)
\tag{5}
$$

where $J$ is rotary inertia, defined as:

$$
J = \begin{bmatrix} I_{xx} & 0 & I_{xz} \\ 0 & I_{yy} & 0 \\ I_{zx} & 0 & I_{zz} \end{bmatrix}
\tag{6}
$$

and $M_a = [l\ m\ n]$ is the aerodynamic moment, defined by:

$$
M_a = \begin{bmatrix} l \\ m \\ n \end{bmatrix} = \bar{q} S \begin{bmatrix} b \cdot C_l \\ \bar{c} \cdot C_m \\ b \cdot C_n \end{bmatrix} = \bar{q} S \begin{bmatrix} b \cdot [(C_{l,base}(\alpha, \beta, V) + \sum_{i=1}^{j} C_{l,i}(\alpha, \beta, \delta))] \\ \bar{c} \cdot [(C_{m,base}(\alpha, \beta, V) + \sum_{i=1}^{j} C_{m,i}(\alpha, \beta, \delta))] \\ b \cdot [(C_{n,base}(\alpha, \beta, V) + \sum_{i=1}^{j} C_{n,i}(\alpha, \beta, \delta))] \end{bmatrix}
\tag{7}
$$

where the $\bar{q}$ is the dynamic pressure, $b$ is span, $\bar{c}$ is the mean aerodynamic chord, $\delta \in \delta_{max}$ represents the deflection of effectors, $\delta_{max}$ is the deflection range of effectors [3], $C_{\cdot,base}(\alpha, \beta, V)$ and $\sum_{i=1}^{j} C_{\cdot,i}(\alpha, \beta, \delta)$ are the aerodynamic coefficients generated by the body and control surfaces.

**Table 2.** Nomenclature of variables.

| Parameter | Nomenclature | Unit |
|---|---|---|
| $F$ | = sum of aerodynamic force and thrust | LBS |
| $l, m, n$ | = aerodynamic rolling, pitching, and yaw moment | LBS·ft |
| $p, q, r$ | = body-axis roll, pitch, and yaw rate | rad/s |
| $V$ | = airspeed | ft/s |
| $\gamma$ | = flight path angle | rad |
| $\chi$ | = flight path sideslip angle | rad |
| $\alpha$ | = angle of attack | rad |
| $\beta$ | = sideslip angle | rad |
| $\mu$ | = bank angle about the velocity vector | rad |

The coordinate system involved in the above kinetic equations is shown in Figure 2. Equation (1) is defined in the tangent-plane coordinate system, which is aligned as a geographic system but has its origin fixed at a point of interest on the spheroid; Equation (2) is defined in the wind-axes system, and Equation (5) is defined in the body-fixed coordinate system. The relationship between the wind-axes system and the body-fixed coordinate system is shown in Figure 1. The origin of both is at the UAV's center of gravity, but the X-axis of the body-fixed coordinate system points in the direction of the nose, and the X-axis of the wind-axes system points in the direction of the relative wind [53].

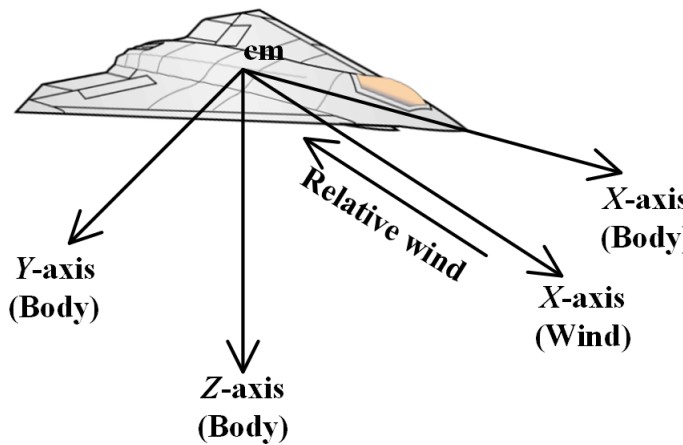

**Figure 2.** Explanation of wind-axes system and body-fixed coordinate system.

To facilitate the design of the attitude tracker, we first construct a control-oriented model that considers disturbances and model uncertainty caused by inaccurate aerodynamic parameters. Define $x_1 = [\mu\ \alpha\ \beta]$, $x_2 = [p\ q\ r]$, then

$$\begin{cases} \dot{x}_1 = f_1(x_1) + g_1(x_1)x_2 \\ \dot{x}_2 = f_2(x_2) + g_2(x_1, \delta) + d_t \end{cases} \tag{8}$$

where

$$f_1(x_1) = \begin{bmatrix} c_\alpha c_\beta & 0 & s_\alpha \\ s_\beta & 1 & 0 \\ s_\alpha c_\beta & 0 & -c_\alpha \end{bmatrix}^{-1} \cdot -T_{bv}^T \begin{bmatrix} -\dot{\chi}s_\gamma \\ \dot{\gamma} \\ \dot{\chi}c_\gamma \end{bmatrix} \tag{9}$$

$$g_1(x_1) = \begin{bmatrix} c_\alpha c_\beta & 0 & s_\alpha \\ s_\beta & 1 & 0 \\ s_\alpha c_\beta & 0 & -c_\alpha \end{bmatrix}^{-1} \tag{10}$$

$$f_2(x_2) = J^{-1}x_2 \times Jx_2 \tag{11}$$

$$g_2(x_1, \delta) = J^{-1}M_a - g_{2,e}(x_1, \delta) \tag{12}$$

$$d_t = g_{2,e}(x_1, \delta) + d \tag{13}$$

where $d_t$ stand for total uncertainty, $g_{2,e}$ represents the control effectiveness that aerodynamic data fail to curve, and $d$ represents the external disturbances.

## 3. Problem Formulation

The control structure is shown in Figure 3, donating $\cdot^c$ as the command signal. The control system consists of two parts, attitude control using NDI and angular rate control using ADP. The attitude control is to give the proper angular command so that the UAV can track the attitude command, assuming that the derivative of the attitude command is known, and gives the angular rate command as [44]:

$$x_2^c = g_1^{-1}(x_1) \cdot (\dot{x}_1^d - f_1(x_1)) \tag{14}$$

where $x_2^c = [p_c, q_c, r_c]^T$.

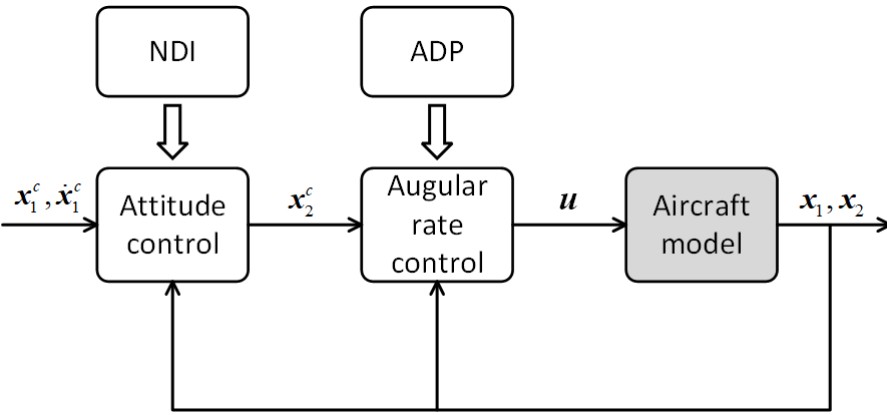

**Figure 3.** Control structure.

The goal of angular rate control is to give effectors' deflection so that the angular rate command can be tracked optimally according to the performance function. First, we constructed an augmented system:

$$\dot{x}_2 = f_2(x_2) + g_2(x_1, \delta) \tag{15}$$

$$\dot{x}_2^d = \lambda(x_2^c - x_2) \tag{16}$$

where $\lambda$ is positively defined, and $x_2^d$ represents the desired angular rate.

Define $X(t) = [x_2^T, x_2^{d^T}]^T$, and rewrite the augmented system in the compact form:

$$\dot{X} = F(X) + G(X, \delta) + C + D \tag{17}$$

where $F(X) = \begin{bmatrix} f_2(x_2) \\ -\lambda x_2 \end{bmatrix}$, $G(X) = \begin{bmatrix} g_2(x_1, \delta) \\ 0_{3 \times 1} \end{bmatrix}$, $C = \begin{bmatrix} 0_{3 \times 1} \\ \lambda x_2^c \end{bmatrix}$, $D = \begin{bmatrix} d_t \\ 0_{3 \times 1} \end{bmatrix}$.

**Remark 2.** *From Equation (14), it could be known that it is difficult to obtain the dynamic of $x_2^c$ because of its dependence on the second order derivative of $x_1^d$. In many control methods, such as incremental control methods [14], $\dot{x}_1^c$ is readily available, but $\dot{x}_2^c$ is usually obtained through digital differentiation, which is not only sensitive to noise but also increases the computational complexity. Therefore, in ADP, the dynamic of $x_2^d$ is constructed using NDI, and the use of $\ddot{x}_1^d$ could be avoided in this way.*

To facilitate the following analysis, the following assumptions are given:

**Assumption 1.** *The total model uncertainty is bounded, there exist $D_{max} > 0$, let $\|D\| < D_{max}$.*

**Assumption 2.** *$\frac{\partial g_2(x_1, \delta)}{\partial \delta} \in \mathbb{R}_{3 \times 11}$ is line nonsingular, i.e., for $\|D\| < D_{max}$, there exist $|\Delta \delta| < \bar{\delta}$, $\bar{\delta} > 0$ such that $\frac{\partial g_2(x_1, \delta)}{\partial \delta} \Delta \delta = D$.*

**Remark 3.** *From Equation (8), it is obvious that the total model uncertainties can be equated to time-varying aerodynamic moment disturbances. To put it bluntly, Assumption 2 means that such disturbances are contained in the attainable moments set of ICE, and considering the redundant effectors that ICE is equipped with, its attainable moments set [54] has already been greatly expanded. From Equation (13), it also could be found that $g_2(x_1, \delta)$ could only reflect the aerodynamic characteristic of ICE to a certain degree. Except for the accuracy loss in the wind-tunnel test, the raw aerodynamic data also must be well-tailored to make it more suitable for flight control design, and the aerodynamic that $g_2(x_1, \delta)$ failed to reflect are seen as model uncertainties. Therefore, it is reasonable to say that $g_2(x_1, \delta)$ has the properties stated in Assumption 2.*

Specifically, the angular rate control is to ensure the $x_2$ tracks the $x_2^c$ by minimizing the following performance function. Consider that when the command signal does not converge to 0, the control input also tends not to be 0. To ensure the boundness of the performance function, the following discounted performance function is introduced:

$$J(\boldsymbol{X}, \boldsymbol{\delta}) = \int_t^{\infty} e^{-\upsilon(\tau - t)} (\bar{\boldsymbol{\delta}}^T \boldsymbol{R} \bar{\boldsymbol{\delta}} + \boldsymbol{e_2}^T \boldsymbol{Q} \boldsymbol{e_2} + \boldsymbol{\delta}^T \boldsymbol{R} \boldsymbol{\delta}) \, d\tau \tag{18}$$

where $\boldsymbol{e_2} = \boldsymbol{x_2} - \boldsymbol{x_2^d}$, $\boldsymbol{Q}$ and $\boldsymbol{R}$ are positively defined matrices, $\upsilon > 0$ is the discounted factor. Compared to the conventional performance function, the upper bound of model uncertainties $\bar{\boldsymbol{\delta}}$ is introduced into the performance function.

**Remark 4.** *As a tried-and-tested control algorithm, NDI has been used in flight control for decades. In NDI, by feeding back the error between $x_2$ and $x_2^d$, the tracking of $x_2^c$ could be achieved. Therefore, in the above performance function design, we also use the feedback of $x_2^d$ instead of $x_2^c$.*

Since the exact value of model uncertainties is unavailable, only the optimal tracker for the following nominal system that excludes the model uncertainties can be obtained:

$$\dot{\boldsymbol{X}} = \boldsymbol{F}(\boldsymbol{X}) + \boldsymbol{G}(\boldsymbol{X}, \boldsymbol{\delta}) + \boldsymbol{C} \tag{19}$$

However, the optimal tracker designed for the nominal system using the performance function in Equation (18) is capable of handling model uncertainties. This point will be elaborated on later, and the optimal tracker for the nominal system is derived as follows:

Define $\boldsymbol{Q_T} = [\boldsymbol{I_{3 \times 3}}, \ -\boldsymbol{I_{3 \times 3}}]^T \boldsymbol{Q} [\boldsymbol{I_{3 \times 3}}, \ -\boldsymbol{I_{3 \times 3}}]$, then the discounted performance function could be modified as:

$$J(\boldsymbol{X}, \boldsymbol{\delta}) = \int_t^{\infty} e^{-\upsilon(\tau - t)} (\bar{\boldsymbol{\delta}}^T \boldsymbol{R} \bar{\boldsymbol{\delta}} + \boldsymbol{X}^T \boldsymbol{Q_T} \boldsymbol{X} + \boldsymbol{\delta}^T \boldsymbol{R} \boldsymbol{\delta}) \, d\tau \tag{20}$$

Then define the optimal value function:

$$V(\boldsymbol{X}) = \min J(\boldsymbol{X}, \boldsymbol{\delta}) \tag{21}$$

**Remark 5.** *It is worth noting that the constraints imposed by the effector are not taken into account when solving the optimal control problem. However, it is possible to effectively prevent effector saturation by adjusting the weights in the performance function. This approach is widely adopted in the solution of optimization problems.*

Differentiating Equation (20) and noting Equation (21) give the following Hamilton–Jacobi–Bellman (HJB) equation:

$$H(V, \boldsymbol{\delta}) \triangleq \bar{\boldsymbol{\delta}}^T \boldsymbol{R} \bar{\boldsymbol{\delta}} + \boldsymbol{X}^T \boldsymbol{Q_T} \boldsymbol{X} + \boldsymbol{\delta}^T \boldsymbol{R} \boldsymbol{\delta} - \upsilon V + V_X^T (\boldsymbol{F}(\boldsymbol{X}) + \boldsymbol{G}(\boldsymbol{X}, \boldsymbol{\delta}) + \boldsymbol{C}) = 0 \tag{22}$$

where $V_X = \frac{\partial V}{\partial X}$. Applying stationarity condition $\partial H(V, \boldsymbol{\delta}) / \partial \boldsymbol{\delta} = 0$, then we have the optimal tracker:

$$\boldsymbol{\delta}^* = -\frac{1}{2} \boldsymbol{R}^{-1} \boldsymbol{G_\delta}^T V_X \tag{23}$$

where $\boldsymbol{G_\delta} = \frac{\partial \boldsymbol{G}(\boldsymbol{X}, \boldsymbol{\delta})}{\partial \boldsymbol{\delta}}$. If the HJB equation is solved, the optimal control can be obtained. Therefore, the main issue of this paper is to solve the HJB equation using ADP.

**Remark 6.** *From Equation (23), it can be found that the only difference between the proposed optimal control for multi-input nonaffine systems and traditional optimal control lies in the use of $\boldsymbol{G_\delta^T}$, which is obtained through digital differentiation of aerodynamic data. In the theoretical study of the optimal control of nonaffine systems [49,50,55–57], the nonaffine part of the system is usually treated as completely unknown. Therefore, complex model identification methods are*

*needed in these studies. However, for realistic control systems, even if the accurate analytical model is not available, the data-based model can be built by observing the input and output of the system. With the improvement of wind-tunnel tests for UAV modeling, the aerodynamic data we obtain is more accurate than ever. The aforementioned theoretical studies, however, have not made sufficient use of these data. This is undoubtedly a huge waste. In this paper, the wind-tunnel data are used to assist ADP. By introducing $\mathbf{D_{max}}$ into the performance function, the ADP shows robustness against uncertainties in wind-tunnel data. Compared with online model identification, many more computational resources could be saved in this way.*

The following lemma shows that with the performance function Equation (20), the optimal tracker in Equation (23) shows robustness against model uncertainties.

**Lemma 1.** *Assume that the optimal control δ\* of the nominal system (19) with performance function (20) exist, δ\* could also make the system with model uncertainties (17) asymptotically stable.*

**Proof.** To facilitate the proof, an auxiliary system is proposed:

$$\begin{cases} \dot{\mathbf{X}} = \mathbf{F}(\mathbf{X}) + \mathbf{G}(\mathbf{X}, \boldsymbol{\delta}) + \mathbf{C} + \mathbf{D} \\ \dot{a} = -\dfrac{1}{2}va \end{cases} \tag{24}$$

It is obvious that the subsystem $\dot{a} = -\frac{1}{2}va$ is asymptotically stable. Therefore, as long as the auxiliary system is proven to be asymptotically stable, Lemma 1 holds. Consider the optimal value function in Equation (21), for all $\mathbf{X} \neq 0$, $V > 0$ and $V = 0$ only when $\mathbf{X} = 0$. Therefore, choose the Lyapunov function as $V_R = a^2 V$, and the time derivation of $V_R$ is:

$$\begin{aligned} \dot{V}_R &= a^2 \dot{V} - a^2 v V \\ &= a^2 [V_X^T(\mathbf{F}(\mathbf{X}) + \mathbf{G}(\mathbf{X}, \boldsymbol{\delta}) + \mathbf{C}) + V_X^T \mathbf{D} - vV] \end{aligned}$$

according to the HJB equation and stationarity condition, we have:

$$\dot{V}_R = a^2 [-\bar{\boldsymbol{\delta}}^T \mathbf{R} \bar{\boldsymbol{\delta}} - \mathbf{X}^T \mathbf{Q}_T \mathbf{X} - \boldsymbol{\delta}^T \mathbf{R} \boldsymbol{\delta} + vV - 2\boldsymbol{\delta}^T \mathbf{R}^{\frac{1}{2}} \cdot \mathbf{R}^{\frac{1}{2}} \mathbf{G}_\delta^+ \mathbf{D} - vV]$$

where the $\mathbf{G}_\delta^+$ is the Moore-Penrose inverse of $\mathbf{G}_\delta$. According to Assumption 2 $\mathbf{G}_\delta^+ \mathbf{D} = \Delta \boldsymbol{\delta}$, such that:

$$\begin{aligned} \dot{V}_R &= a^2 [-\bar{\boldsymbol{\delta}}^T \mathbf{R} \bar{\boldsymbol{\delta}} + \Delta \boldsymbol{\delta}^T \mathbf{R} \Delta \boldsymbol{\delta} - \mathbf{X}^T \mathbf{Q}_T \mathbf{X} - \boldsymbol{\delta}^T \mathbf{R} \boldsymbol{\delta} - 2\boldsymbol{\delta}^T \mathbf{R}^{\frac{1}{2}} \cdot \mathbf{R}^{\frac{1}{2}} \Delta \boldsymbol{\delta} - \Delta \boldsymbol{\delta}^T \mathbf{R} \Delta \boldsymbol{\delta}] \\ &= a^2 [-\bar{\boldsymbol{\delta}}^T \mathbf{R} \bar{\boldsymbol{\delta}} + \Delta \boldsymbol{\delta}^T \mathbf{R} \Delta \boldsymbol{\delta} - \mathbf{X}^T \mathbf{Q}_T \mathbf{X} - (\mathbf{R}^{\frac{1}{2}} \boldsymbol{\delta} + \mathbf{R}^{\frac{1}{2}} \Delta \boldsymbol{\delta})^T (\mathbf{R}^{\frac{1}{2}} \boldsymbol{\delta} + \mathbf{R}^{\frac{1}{2}} \Delta \boldsymbol{\delta})] \end{aligned} \tag{25}$$

according to Assumption 2, $|\Delta \boldsymbol{\delta}| < \bar{\boldsymbol{\delta}}$ and $\mathbf{R}$ is positively definite, $-\bar{\boldsymbol{\delta}}^T \mathbf{R} \bar{\boldsymbol{\delta}} + \Delta \boldsymbol{\delta}^T \mathbf{R} \Delta \boldsymbol{\delta} < 0$. Therefore, it is clear that $\dot{V}_R$ is negative definite, the auxiliary system is asymptotically stable, and Lemma 1 is proven.

## 4. Main Result

As mentioned above, the main issue of this paper is solving the HJB equation and obtaining the optimal value function. The following single-layer neural network(NN) is applied to approximate the optimal value function:

$$V(\mathbf{X}) = \mathbf{W}^T \mathbf{\Psi}(\mathbf{X}) + \varsigma \tag{26}$$

where $\mathbf{\Psi}(\mathbf{X}) \in \mathbb{R}^l$ is the activation function vector, $l$ is the number of neurons, $\varsigma$ is the approximation error, and the derivative to state of $V(\mathbf{X})$ is:

$$V_X = \nabla \mathbf{\Psi} \mathbf{W} + \nabla \varsigma \tag{27}$$

where $\mathbf{W} \in \mathbb{R}^l$ is unknown ideal weights, $\nabla \mathbf{\Psi} = \frac{\partial \mathbf{\Psi}^T}{\partial \mathbf{X}}$, $\nabla \varsigma = \frac{\partial \varsigma}{\partial \mathbf{X}}$.

Suppose that the optimal value function is continuous and is defined on a bounded closed interval, according to the Weierstrass approximation theorem [58]. As $l$ increases, the optimal value function can be uniformly approximated by the NN with arbitrarily high preciseness, which means that $\varsigma$ and $\nabla \varsigma$ can be arbitrarily small. In practice, we use a critical NN to approximate the optimal value function:

$$\hat{V}(X) = \hat{W}^T \Psi(X) \tag{28}$$

where $\hat{W}$ is the estimation of unknown weights $W$. Then the near-optimal control $\hat{\delta}$ can be obtained:

$$\hat{\delta} = -\frac{1}{2} R^{-1} G_\delta^T \nabla \Psi \hat{W} \tag{29}$$

In this paper, ADP updates the critical NN online using sampling data. As more sampling data ADP obtains, $\hat{W}$ would approach $W$ gradually. The rest of this section contains two parts. The first part introduces the update law of critical NN. The second part is stability analysis, in which we will discuss why the $\hat{W}$ could approach $\hat{W}$ and why the system stays stable.

### 4.1. Update Law for Critical NN

The goal of the update scheme is to minimize $\tilde{W} = W - \hat{W}$ the estimation error of unknown NN weights. To derive the update law, substitute Equations (26) and (27) into HJB Equation (22), we have:

$$\begin{aligned}
0 = H(V, \delta) &= \bar{\delta}^T R \bar{\delta} + X^T Q_T X + \delta^T R \delta - v(W^T \Psi(X) + \varsigma) \\
&\quad + (W^T \nabla \Psi^T + \nabla \varsigma^T)(F(X) + G(X, \delta) + C) \\
&= \Lambda + \hat{W}^T Y + \tilde{W}^T Y + \varsigma_H
\end{aligned} \tag{30}$$

where

$$\Lambda = \bar{\delta}^T R \bar{\delta} + X^T Q_T X + \delta^T R \delta \tag{31}$$

$$Y = -v\Psi(X) + \nabla \Psi^T(F(X) + G(X, \delta) + C) \tag{32}$$

$$\varsigma_H = -v\varsigma + \nabla \varsigma^T(F(X) + G(X, \delta) + C) \tag{33}$$

From Equation (30), both the estimation error $\tilde{W}$ and the weights of critical NN $\hat{W}$ appear linearly. As discussed in Section 2, $\varsigma_H$ is bounded and tends to be zero as $l$ increases. Moreover, the rest of variables $Y, \Lambda$ are accessible. This allows us to design the update law that minimizes the estimation error. Therefore, we define the following filter:

$$\begin{cases} \dot{P} = -kP + YY^T, P(0) = 0 \\ \dot{Q} = -kQ + Y\Lambda, Q(0) = 0 \end{cases} \tag{34}$$

where $k \in \mathbb{R}^+$. The solution of the filter is:

$$\begin{cases} P = \int_0^t e^{-k(t-\tau)} YY^T \, d\tau \\ Q = \int_0^t e^{-k(t-\tau)} Y\Lambda \, d\tau \end{cases} \tag{35}$$

According to Equations (30) and (35), we have:

$$0 = Q + P\hat{W} + P\tilde{W} + \mu \tag{36}$$

where $\mu = \int_0^t e^{-k(t-\tau)} \mathbf{Y}\varsigma_H \, d\tau$. According to the above analysis, with sufficient neurons, $\varsigma \to 0$ and $\nabla\varsigma \to 0$. Therefore, it is reasonable to find that $\mu$ is bounded, i.e., $|\mu| < \bar{\mu}$, where $\bar{\mu} \in \mathbb{R}^+$.

Define an auxiliary vector $\mathbf{M} \in \mathbb{R}^l$:

$$\mathbf{M} = \mathbf{Q} + \mathbf{P}\hat{W} \tag{37}$$

According to Equation (36), $\mathbf{M}$ could be seen as the estimation error of NN weights:

$$\mathbf{M} = -\mathbf{P}\tilde{W} - \mu \tag{38}$$

Hence, we could obtain the online update law of $\hat{W}$ as:

$$\dot{\hat{W}} = -K\mathbf{M} \tag{39}$$

where $K$ is constant positively defined matrix.

The update law in this paper is fundamentally different from some of the current methods [41,42] that employ the gradient-based algorithm to minimize the bellman error and only could ensure ultimately uniform boundlessness (UUB). The update law used here intends to use measurable state values and the weights of critical NN to represent the unknown estimation error of NN weights. In this way, the estimation error of NN weights can be ensured to be asymptotically convergent, and good convergence makes this update rate more suitable for online control systems. In what follows, we will show that $\hat{W}$ could converge to the domain of $W$.

**Remark 7.** *The idea of this paper is to use the optimal tracker of nominal systems with modified performance functions to handle the systems with disturbances and model uncertainties. This idea can be well applied in linear systems [29] since the optimal control for linear nominal systems is easy to obtain by solving the Recatii equation. For the nonlinear system, however, it needs to solve the HJB equation to obtain the optimal control. For algorithms such as ADP that solve the HJB equation online, it gives the approximate solution of the HJB equation based on the sampled data. Still, for the systems that suffer from disturbance, it is impossible to measure the state value of the nominal system. According to Equations (31), (32) and (34), the information that update law uses include the quadratic function of effector deflection $\delta^T R \delta$, the nominal system dynamic $F(X) + G(X, \delta) + C$, and the quadratic function of state value $X^T Q_T X$. The $\delta^T R \delta$ and $F(X) + G(X, \delta) + C$ are directly accessible. Only the $X^T Q_T X$ is affected by disturbances and may influence the update law.*

*To address this problem, the filter system in Equation (34) is used here. Instead of using these values directly, the values used by the update law are processed by the filter system. In this way, the influence of disturbance on $X^T Q_T X$ could be mitigated to a certain extent and make the update law more applicable. To further illustrate this opinion, the following example is introduced:*

*For the system with disturbance:*

$$\begin{cases} \dot{x}_1 = -x_1 + x_2 + d \\ \dot{x}_2 = x_3 \\ \dot{x}_3 = -x_2 \end{cases} \tag{40}$$

*where d is the disturbance.*

*Figure 4 shows the unfiltered value of $x_1^2$, and the value of $x_1^2$ filtered by $\frac{1}{0.8s+1}$. The real system is affected by disturbance, while the nominal system is not. From Figure 4a, there is a distinct difference between the unfiltered $x_1^2$ of the real system and the nominal system. But by filtering the two signals, the difference between the two becomes significantly smaller, as shown in Figure 4b. By choosing the parameters of the filter wisely, the filtered $x_1^2$ of the real system could approximate the $x_1^2$ of the nominal system pretty well.*

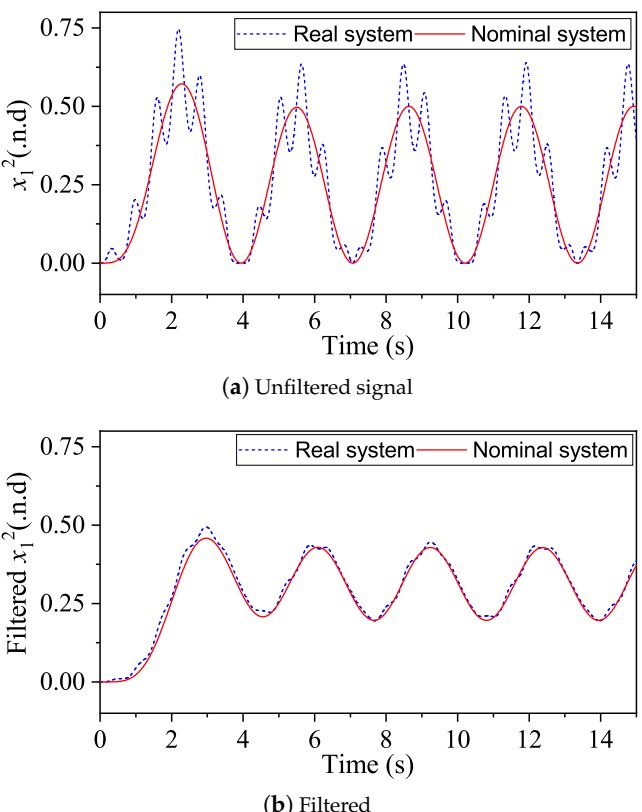

**(a)** Unfiltered signal

**(b)** Filtered

**Figure 4.** Effectiveness of the filter.

It should be noted that since the filter used in this paper is only the simplest first-order low-pass filter, this method could achieve quite a good result when the frequency of disturbances is higher than the frequency of state. Considering that high-frequency disturbance is a common kind of disturbance in control systems, the applicability of this method is acceptable. To cope with more complex situations, more targeted filters can be used according to the knowledge of the disturbance.

### 4.2. Stability Analysis

To prove that our approach can solve the optimal control online while making the tracking error converge, the following stability analysis is performed. Accordingly, modify the nominal system (24) in the following closed-loop form:

$$\begin{cases} \dot{X} = F(X) + G(X, \delta^*) + C + G(X, \hat{\delta}) - G(X, \delta^*) \\ \dot{a} = -\dfrac{1}{2}va \end{cases} \tag{41}$$

**Assumption 3.** *There exist constants $\bar{f}, \bar{g} \in \mathbb{R}^+$ such that $\|F(X)\| \leqslant \bar{f}\|X\|$, $\|G(X, \delta)\| \leqslant \bar{g}\|X\|$.*

**Assumption 4.** *The $G(X, \delta)$ is Lipschitz continuous with respect to $\delta$, i.e., $\|G(X, \delta_1) - G(X, \delta_2)\| \leqslant L\|\delta_1 - \delta_2\|$, where $L \in \mathbb{R}^+$.*

**Assumption 5.** *The command signal is bounded, i.e., there exists a constant $\bar{c} \in \mathbb{R}^+$ such that $\|C\| \leqslant \bar{c}$.*

**Assumption 6.** *The activation function vector $\Psi(X)$ and its derivative $\nabla\Psi$ is bounded.*

**Lemma 2.** *For the PE condition is satisfied for the regressor $Y$ of NN, the optimal control and the update law could stabilize the tracking error of the nominal system (19), and the near-optimal*

*control converges to a bounded neighborhood around optimal control, i.e., $\boldsymbol{X}^T\boldsymbol{Q_T}\boldsymbol{X} \leqslant \xi_X$ and $\|\hat{\boldsymbol{\delta}} - \boldsymbol{\delta}^*\| \leqslant \xi_\delta$, where $\xi_X$, $\xi_\delta \in \mathbb{R}^+$.*

**Proof.** The Lyapunov function is constructed as:

$$J = J_1 + J_2 \tag{42}$$

where $J_1 = \frac{1}{2}\tilde{\boldsymbol{W}}^T\boldsymbol{K}^{-1}\tilde{\boldsymbol{W}}$, $J_2 = \zeta_1\boldsymbol{X}^T\boldsymbol{Q_T}\boldsymbol{X} + \zeta_2 a^2 V$.

According to the Lemma 1 in [37], if the PE condition is satisfied for the regressor $\boldsymbol{Y}$ of NN, then $\boldsymbol{P}$ is positively defined, and since the inequality $2ab \leqslant \eta a^2 + \frac{b^2}{\eta}$, the derivative of $J_1$ is:

$$
\begin{aligned}
\dot{J}_1 &= -\tilde{\boldsymbol{W}}^T\boldsymbol{K}\tilde{\boldsymbol{W}} + \tilde{\boldsymbol{W}}^T\boldsymbol{\mu} \\
&\leqslant -(\lambda_{min}(\boldsymbol{P}) - \eta_1)\|\tilde{W}\|^2 + \frac{1}{\eta_1}\|\boldsymbol{\mu}\| \\
&\leqslant -(\lambda_{min}(\boldsymbol{P}) - \eta_1)\|\tilde{W}\|^2 + \frac{1}{\eta_1}\bar{\mu}^2
\end{aligned}
\tag{43}
$$

where $\lambda_{min}(\boldsymbol{P})$ represents minimum eigenvalue of $\boldsymbol{P}$, and $\eta_1 \in \mathbb{R}^+$. According to Equations (22) and (24), the derivative of $J_2$ is:

$$
\begin{aligned}
\dot{J}_2 &= 2\zeta_1\boldsymbol{X}^T\boldsymbol{Q_T}\dot{\boldsymbol{X}} - va^2V + a^2\dot{V} \\
&= 2\zeta_1\boldsymbol{X}^T\boldsymbol{Q_T}[\boldsymbol{F}(\boldsymbol{X}) + \boldsymbol{G}(\boldsymbol{X},\boldsymbol{\delta}^*) + \boldsymbol{C} + \boldsymbol{G}(\boldsymbol{X},\hat{\boldsymbol{\delta}}) - \boldsymbol{G}(\boldsymbol{X},\boldsymbol{\delta}^*)] - va^2V \\
&\quad + a^2(-\bar{\boldsymbol{\delta}}^T\boldsymbol{R}\bar{\boldsymbol{\delta}} - \boldsymbol{X}^T\boldsymbol{Q_T}\boldsymbol{X} - \boldsymbol{\delta}^T\boldsymbol{R}\boldsymbol{\delta} + vV) \\
&= 2\zeta_1\boldsymbol{X}^T\boldsymbol{Q_T}\boldsymbol{F}(\boldsymbol{X}) + 2\zeta_1\boldsymbol{X}^T\boldsymbol{Q_T}\boldsymbol{G}(\boldsymbol{X},\boldsymbol{\delta}^*) + 2\zeta_1\boldsymbol{X}^T\boldsymbol{Q_T}\boldsymbol{C} + 2\zeta_1\boldsymbol{X}^T\boldsymbol{Q_T}(\boldsymbol{G}(\boldsymbol{X},\hat{\boldsymbol{\delta}}) - \boldsymbol{G}(\boldsymbol{X},\boldsymbol{\delta}^*)) \\
&\quad - \zeta_2 a^2\bar{\boldsymbol{\delta}}^T\boldsymbol{R}\bar{\boldsymbol{\delta}} - \zeta_2 a^2\boldsymbol{X}^T\boldsymbol{Q_T}\boldsymbol{X} - \zeta_2 a^2\boldsymbol{\delta}^T\boldsymbol{R}\boldsymbol{\delta} \\
&\leqslant \zeta_1(\eta_2\|\boldsymbol{X}^T\boldsymbol{Q_T}\|^2 + \frac{1}{\eta_2}\|\boldsymbol{F}(\boldsymbol{X})\|^2) + \zeta_1(\eta_3\|\boldsymbol{X}^T\boldsymbol{Q_T}\|^2 + \frac{1}{\eta_3}\|\boldsymbol{G}(\boldsymbol{X},\boldsymbol{\delta}^*)\|^2) \\
&\quad + \zeta_1(\eta_4\|\boldsymbol{X}^T\boldsymbol{Q_T}\|^2 + \frac{1}{\eta_4}\|\boldsymbol{C}\|^2) \\
&\quad + \zeta_1(\eta_5\|\boldsymbol{X}^T\boldsymbol{Q_T}\|^2 + \frac{1}{\eta_5}\|(\boldsymbol{G}(\boldsymbol{X},\boldsymbol{\delta}^*) - \boldsymbol{G}(\boldsymbol{X},\hat{\boldsymbol{\delta}}))\|^2) \\
&\quad - \zeta_2 a^2\bar{\boldsymbol{\delta}}^T\boldsymbol{R}\bar{\boldsymbol{\delta}} - \zeta_2 a^2\lambda_{min}(\boldsymbol{Q_T})\|\boldsymbol{X}\|^2 - \zeta_2 a^2\lambda_{min}(\boldsymbol{R})\|\boldsymbol{\delta}^*\|^2
\end{aligned}
\tag{44}
$$

According to Assumptions 3–5, we have:

$$
\begin{aligned}
\dot{J}_2 &\leqslant [(\zeta_1\eta_2 + \zeta_1\eta_3 + \zeta_1\eta_4 + \zeta_1\eta_5)\lambda_{max}^2(\boldsymbol{Q_T}) + \frac{\zeta_1}{\eta_2}\bar{f}^2 + \frac{\zeta_1}{\eta_3}\bar{g}^2 - \zeta_2 a^2\lambda_{min}(\boldsymbol{Q_T})]\|\boldsymbol{X}\|^2 \\
&\quad + \frac{\zeta_1}{\eta_5}L^2 b_w\|\tilde{\boldsymbol{W}}\|^2 + \frac{\zeta_1}{\eta_5}L^2 b_\varsigma\|\nabla\varsigma\|^2 - \zeta_2 a^2\bar{\boldsymbol{\delta}}^T\boldsymbol{R}\bar{\boldsymbol{\delta}} - \zeta_2 a^2\lambda_{min}(\boldsymbol{R})\|\boldsymbol{\delta}^*\|^2
\end{aligned}
\tag{45}
$$

where $b_w = \|\frac{1}{2}\boldsymbol{R}^{-1}\boldsymbol{G}_\delta^T\nabla\boldsymbol{\Psi}\|$, $b_\varsigma = \|\frac{1}{2}\boldsymbol{R}^{-1}\boldsymbol{G}_\delta^T\|$ according to Assumptions 4 and 6, $b_w$ and $b_\varsigma$ are bounded.

Therefore, the derivative of $J$ is:

$$\dot{J} = \dot{J}_1 + \dot{J}_2 \leqslant -\aleph_1\|\boldsymbol{X}\| - \aleph_2\|\tilde{\boldsymbol{W}}\| - \aleph_3\|\boldsymbol{\delta}^*\| + \aleph_4 \tag{46}$$

where

$$
\begin{aligned}
\aleph_1 &= -(\zeta_1\eta_2 + \zeta_1\eta_3 + \zeta_1\eta_4 + \zeta_1\eta_5)\lambda_{max}^2(\boldsymbol{Q_T}) - \frac{\zeta_1}{\eta_2}\bar{f}^2 - \frac{\zeta_1}{\eta_3}\bar{g}^2 + \zeta_2 a^2\lambda_{min}(\boldsymbol{Q_T}) \\
\aleph_2 &= \lambda_{min}(\boldsymbol{P}) - \eta_1 - \frac{\zeta_1}{\eta_5}L^2 b_w \\
\aleph_3 &= \zeta_2 a^2\lambda_{min}(\boldsymbol{R}) \\
\aleph_4 &= +\frac{1}{\eta_1}\bar{\mu}^2 + \frac{\zeta_1}{\eta_5}L^2 b_\varsigma\|\nabla\varsigma\|^2 - \zeta_2 a^2\bar{\boldsymbol{\delta}}^T\boldsymbol{R}\bar{\boldsymbol{\delta}}
\end{aligned}
\tag{47}
$$

By designing the parameters wisely, it could be ensured that $B_1, B_2, B_3 > 0$. $B_4$ is mainly influenced by critical NN's estimation error, which would converge to zero as the number of neurons increases.

According to Equation (47), if the following inequalities hold, $\dot{J}_2$ would be negative defined:

$$\begin{aligned} -\aleph_1 \|X\| + \aleph_4 &< 0 \\ -\aleph_2 \|\tilde{W}\| + \aleph_4 &< 0 \\ -\aleph_3 \|\delta^*\| + \aleph_4 &< 0 \end{aligned} \tag{48}$$

Therefore, according to Lyapunov theory, the closed-loop system is stable, and the weights error of NN $\tilde{W}$ is bounded, consider the estimation error of critical NN is also bounded, the near-optimal control will converge to a bounded neighborhood of optimal control. □

## 5. Simulation Verification

This section presents two representative simulations to illustrate the effectiveness of the ADP-based integrated-control-and-control-allocation scheme. The simulations are conducted using fixed-step ode4(Runge–Kutta) solver. The fixed-step size is 0.01 s. The block diagram of the 6-DOF UAV simulation model can be found in Figure 10 of the report given by Niestroy [7]. Simulation 1 compares our control scheme with conventional incremental dynamic inversion and pseudo-inverse control allocation(INDIPI) to verify the optimality of our method. Simulation 2 aims to test the robustness of the proposed control scheme. It is assumed that the leading-edge actuators are represented by the transfer function $\frac{(18)(100)}{((s+18)(s+100))}$ while all the other actuators, including thrust vectoring, as $\frac{(40)(100)}{((s+40)(s+100))}$.

The initial condition of the UAV is $V(0) = 1240$ ft/s, $\chi(0) = 0$, $\gamma(0) = 0$, $\beta(0) = -0.0196$ deg, $\alpha(0) = 3.759$ deg, $\mu(0) = 0.304$ deg, $p(0) = 0$, $q(0) = 0$, $r(0) = 0$, and the initial height is $H(0) = 10{,}000$ ft. The angle of attack command $\alpha_c$ is generated by a band-limited white noise pass the second order filter $\frac{0.5}{5s^2+2s+0.5}$, the bank angle signal command $\mu_c = 6\sin(0.5t)$, and the sideslip angle command $\beta_c = 0$.

The control parameters are set as $\lambda = 3 \cdot I_{3\times3}$, $R = I_{11\times11}$, $Q = 5 \cdot I_{3\times3}$, $K = 500 \cdot I_{7\times7}$, $v = 1$. Define $e_1 = p - p_c$, $e_2 = q - q_c$, $e_3 = r - r_c$, then the activation function vector are designed as $\Psi(X) = [e_1^2, e_2^2, e_3^2, e_1^2 e_2, e_1 e_2 e_3, e_3^2 e_2, e_2^2 e_3]^T$, with initial weights $\hat{W}(0) = [1000, 1000, 1500, 0, 0, 0, 0]^T$.

**Remark 8.** *We designed the activation function vector as above because it would be easier to find the initial admissible control policy [59]. Clearly, there exists $\frac{\partial e_1^2}{\partial p} = 2e_1$. In this sense, the initial control is equivalent to proportional control. Compared with dealing with complex nonlinear feedback, finding an admissible proportional control law is much easier.*

Model uncertainty exists in both simulations. As mentioned above, model uncertainty mainly comes from inaccurate aerodynamic data, which are used to obtain $G_\delta$. Since the aerodynamic data provided by Niestroy is a series of discrete points, it takes interpolation so that these data are of practical use. In both simulations, different interpolation methods are used for controller design and model construction to simulate that controller cannot access accurate aerodynamic data. Specifically, the cubic spline is applied for model construction, and linear interpolation is used in controller design.

**Remark 9.** *The cubic spline is applied for model construction because the actual aerodynamic data should be continuous and smooth. Meanwhile, using linear interpolation in control could save online computational load. As mentioned in [14], different interpolation methods can cause errors of up to 30%. Take the aerodynamic data of a set of all-moving wingtips as an example, as shown in*

*Figure 5, from which it can be founded that the slopes of the tangents of linear interpolation and cubic spline, i.e.,* tan $\tau_2$ *and* tan $\tau_1$*, are different.*

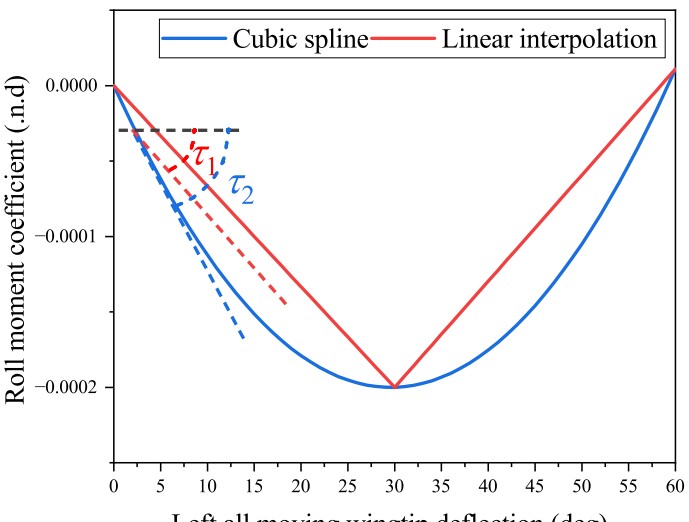

**Figure 5.** Moment coefficients vs. left all-moving tips deflection.

### 5.1. Simulation 1

In this simulation, the ADP-based control scheme is compared with INDIPI [60]. Specifically, our approach and INDIPI are to track the same attitude command, and the performance of both is judged according to the tracking performance, flight quality, and control input. Considering that inaccurate control effectiveness could cause INDIPI to lose stability and this simulation is not to compare the robustness of INDIPI and ADP-based method, INDIPI could obtain accurate model information in this simulation, and model uncertainties only influence our method. Set $\bar{\delta}$ as 11-dimensional vector $[5.6, 5.6, \cdots, 5.6]^T$. Please note that the same NDI scheme is applied in attitude control of the ADP-based method and INDIPI.

The result is shown in Figures 6–17. First, from Figure 17, ADP-based method shows good convergence. Moreover, the proposed method outperforms INDIPI in three ways: superior flight quality, intelligence, and better effector deflection pattern.

From Figures 9–12, the flight quality under ADP control is better than that of INDIPI. From Figure 9, the $p$ signal under ADP control is steadier. After the adjustment period before 5s, the $p$ signal under ADP control keeps steady, while the $p$ signal fluctuates under INDIPI control, such as around 38 s and 44 s. From Figure 10, $q$ chattered all the time under the control of INDIPI, such chattering also can be found in effector deflection as shown in Figure 16, and this can cause fatigue of the effector, which is very dangerous in reality, while the effector deflection under ADP control is more fluent. From Figure 11, it can be found that there is a sudden change in $r$ signal under INDIPI control at 20 s, 29 s, 36 s, and 44 s. It also could be seen that the $r$ signal under ADP control also appears to fluctuate, but, differently from the sudden change under INDIPI control exhibited all the time, it can be found that the fluctuation under ADP control is becoming lighter.

According to the above description, the control performance of our method is better compared to INDIPI. However, our method goes far beyond that. With the help of ADP, our method shows intelligence, i.e., it could improve its policy according to its experience.

The specific manifestations of the ADP-based method's intelligence are $p$ signal under ADP control only fluctuates once around 5 s. After that, it is always very smooth, and, compared with the sudden change in $r$ under INDIPI control exhibited all the time, the fluctuation of $r$ signal under ADP control becomes lighter as the control system runs.

A more extreme example is introduced to further illustrate the intelligence of ADP, as shown in Figure 14, which shows the $p$ signal under the proportional control that adopts the initial weights of critical NN. Comparing Figures 9 and 14 it can be found that no

matter whether under ADP control or proportional control, fluctuation occurred around 5 s. However, such fluctuation only occurs once under ADP control; for proportional control, such fluctuation occurs repeatedly and eventually leads to losing control. From Figure 17, it can be seen that the critical NN weights undergo a large adjustment at 5 s, after which there is no more fluctuation similar to at 5 s. ADP could learn from such fluctuation; therefore, the subsequent policy is more suitable for flight control with a broader flight envelope.

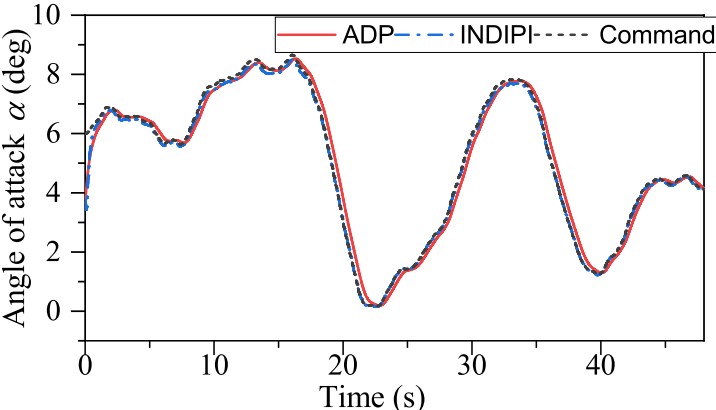

**Figure 6.** Angle of attack.

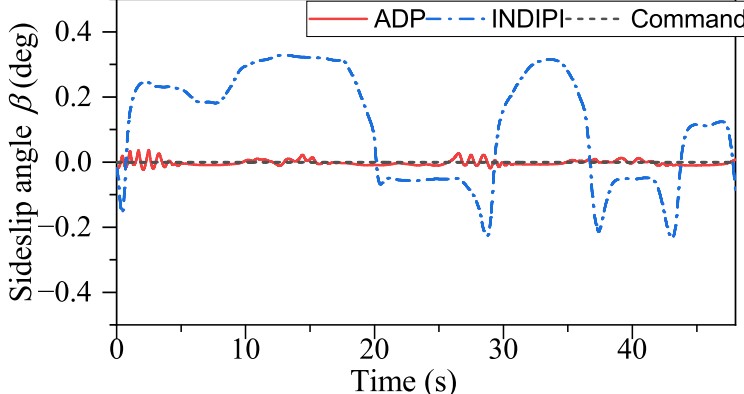

**Figure 7.** Sideslip angle.

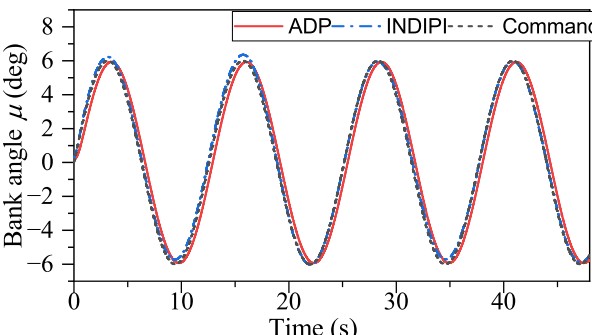

**Figure 8.** Bank angle.

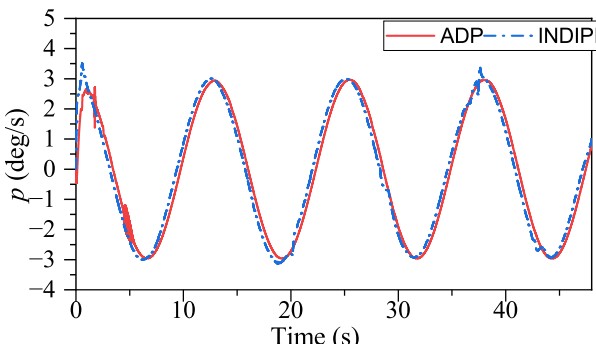

**Figure 9.** Body-axis roll rate.

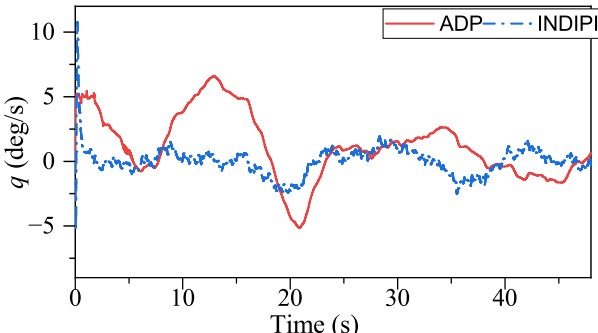

**Figure 10.** Body-axis pitch rate.

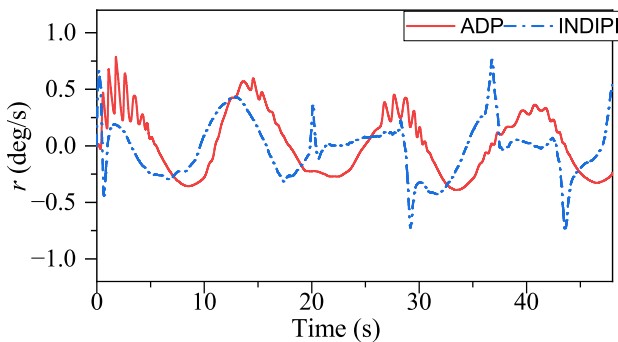

**Figure 11.** Body-axis yaw rate.

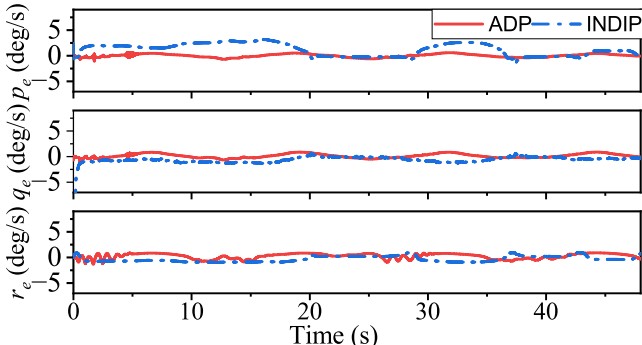

**Figure 12.** Tracking error of $x_2^c$.

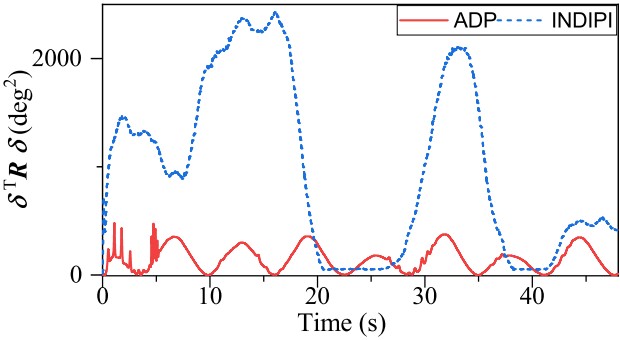

**Figure 13.** Weighted quadratic sum of effector deflection.

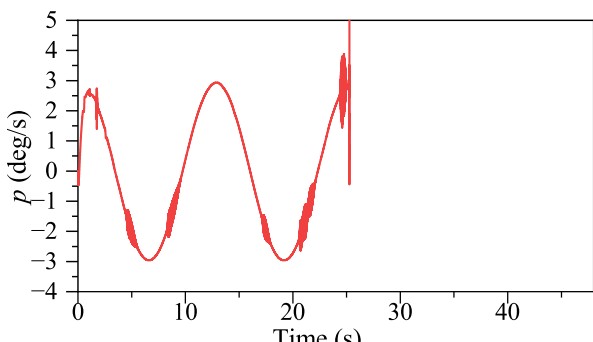

**Figure 14.** $p$ under proportional control.

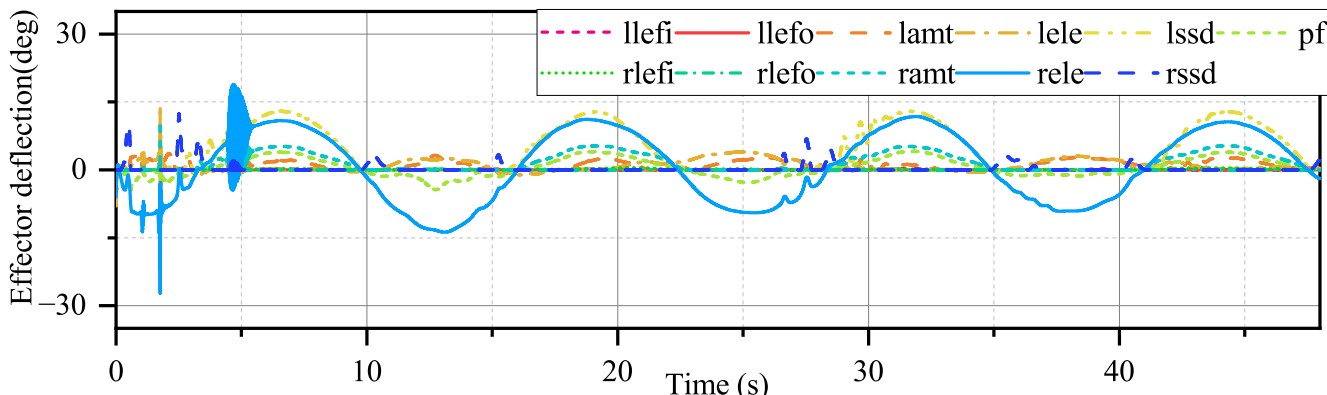

**Figure 15.** Effector deflection of ADP.

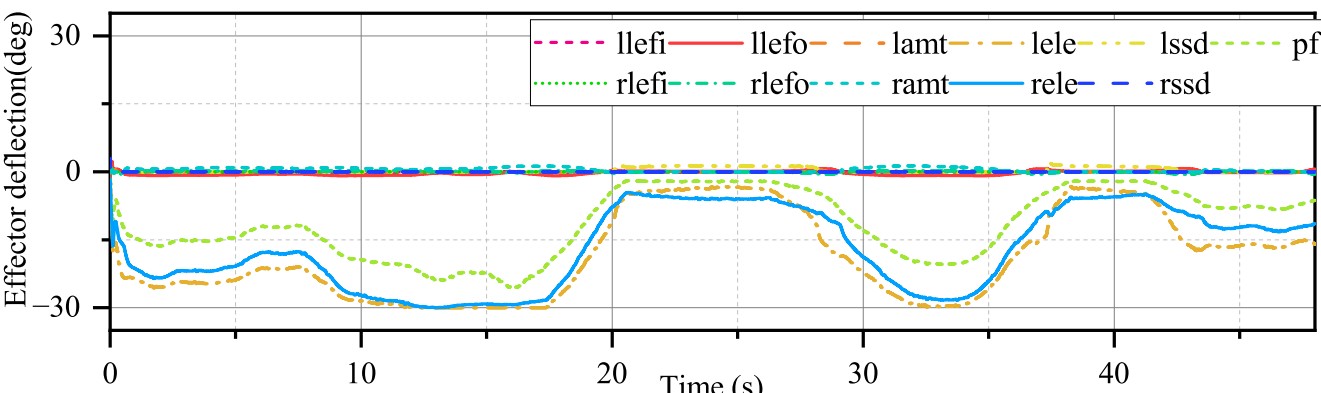

**Figure 16.** Effector deflection of INDIPI.

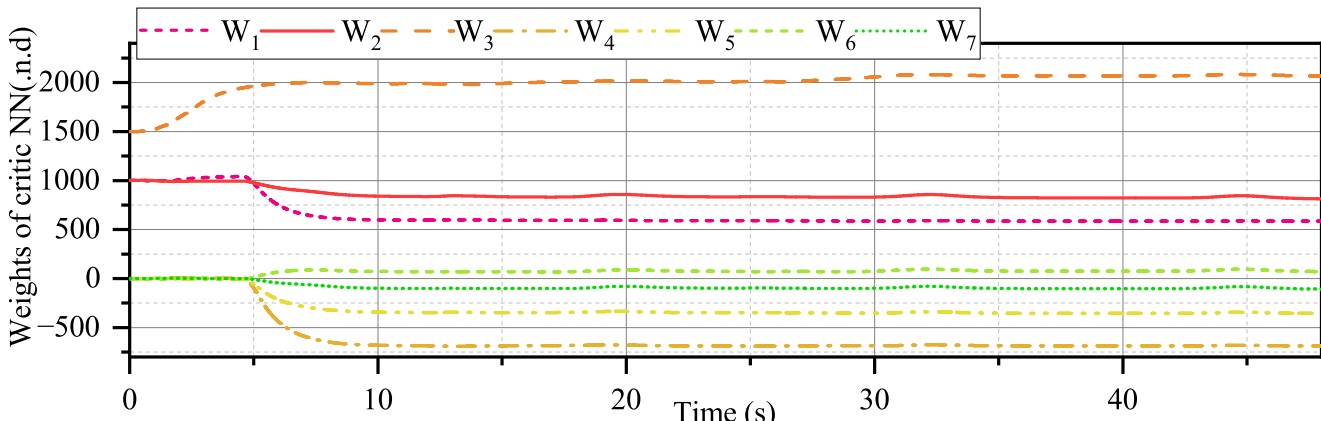

**Figure 17.** Convergence of the critical NN weights.

From Figures 15 and 16, the effector deflection generated by INDIPI and ADP shows different patterns. In Figure 16, it can be seen that only three effectors participate in the process under INDIPI control, even though some effectors appear to be saturated. However, for the ADP, under the modulation of the performance function, more effectors participate in the control process, and the effector deflection amplitude is significantly smaller than that of the INDIPI. From Figure 13, it also can be seen that the weighted quadratic sum of effector deflection given by ADP is much smaller than INDIPI.

Overall, integrated-design ADP's performance is better than conventional INDIPI's. Compared with INDIPI, ADP allows for a trade-off between tracking performance and effector deflection. The performance function dominates such a trade-off, so ADP would not waste too many resources to pursue tiny improvements in tracking performance. Coupling with its learning mechanism, ADP can achieve the same tracking performance as INDIPI in an optimal manner.

*5.2. Simulation 2*

Simulation 2 discusses the robustness of the proposed method and the effect of $\bar{\delta}$. The UAV suffers from aforementioned model uncertainties and external disturbances $d = [0.06\sin(20t), 0.04\cos(20t), 0.03\sin(20t)]^T$. The UAV is to follow the same attitude command as Simulation 1 and set $\bar{\delta}$ as $[17.8, 17.8, \cdots, 17.8]^T$, the result is shown in Figures 18–25.

With the help of ADP, the tracking performance of our method in the presence of external disturbances is unaffected, as shown in Figures 18–20. However, some small chattering can be observed in the angular rate signal, as depicted in Figures 21–23, which is typical for a UAV subject to external disturbance. Nevertheless, this does not compromise the stability of the closed-loop system. Figure 24 demonstrates that more effectors are involved in controlling external disturbances. Most importantly, the convergence of critical NN weights remains satisfactory, as demonstrated by Figure 25.

From the stability analysis, it can be found that our method's robustness comes from $\bar{\delta}$. In the following, the performance of different $\bar{\delta}$ is tested.

We begin by testing $\bar{\delta} = [5.6, 5.6, \cdots, 5.6]^T$. Due to space limitations, we only present the convergence of the critical NN weights in Figure 26. As shown in Figure 26, the convergence of the critical NN in this result is initially similar to Simulation 1. However, the critical NN weights do not ultimately converge due to external disturbances.

From a theoretical perspective, Lemma 1 can explain the non-convergence of the algorithm, as the closed-loop system's stability can only be guaranteed when $\bar{\delta}$ is of sufficient magnitude.

From the other point of view, if the UAV experiences intense external disturbances, the initial sampled data may not provide enough information for ADP to update critical NN weights. This is particularly true when the UAV is required to track random commands, as

the old critical neural network weights may not be equipped to handle new, unforeseen scenarios. As a result, the NN weights may take longer to converge, making it difficult to maintain control of the UAV. In this sense, $\bar{\delta}^T R \bar{\delta}$ not only acts as a means of compensation for external disturbances that may initially affect performance but can also be seen as an estimation of the potential impact of such disturbances on performance function. This helps the ADP better understand the current situation, allowing the weights of the critical NN to converge more quickly to a stable value.

Unlike the affine systems, where the upper bound on the effect of external disturbances on the performance function is easily ascertained [29], for the nonaffine system, the design of $\bar{\delta}$ is more rely on the experience. Still, considering that $\bar{\delta}$ has the actual physical definition, it would not be too hard to find a proper $\bar{\delta}$.

The above result shows that $\bar{\delta}$ must exceed the upper limit of external disturbance effects so that ADP can show robustness.

However, $\bar{\delta}$ should not exceed a reasonable value either. To illustrate this point, we conducted a convergence test of the critical neural network with $\bar{\delta} = [56.4, 56.4, \cdots, 56.4]^T$, which is very large. The result is depicted in Figure 27. It can be observed that the system experiences a significant shift in critical NN weights, leading to a collapse within 3 s. From the analysis of Equation (34), having a too large $\bar{\delta}$ can result in **P** not being positively definite. More bluntly, too large $\bar{\delta}$ could dominate the dynamics of critical neural network weights, causing the ignored of sampled data that aid in policy improvement.

In conclusion, Simulation 2 demonstrates that our method can effectively withstand model uncertainties and external disturbances, given the appropriate selection of $\bar{\delta}$.

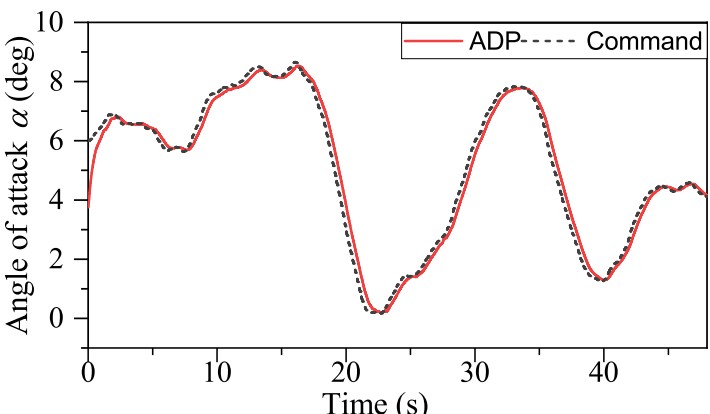

**Figure 18.** Angle of attack.

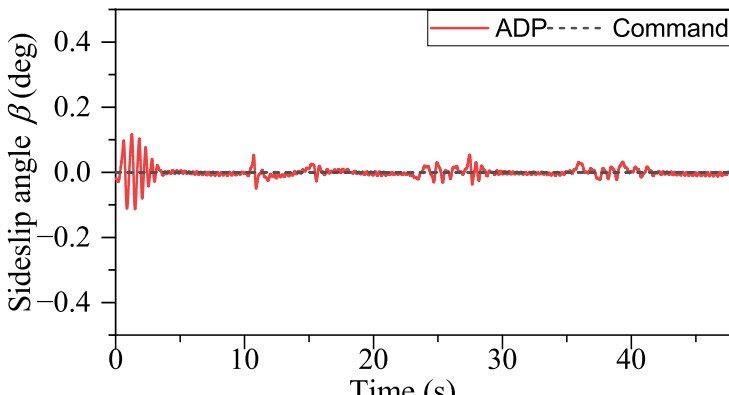

**Figure 19.** Sideslip angle.

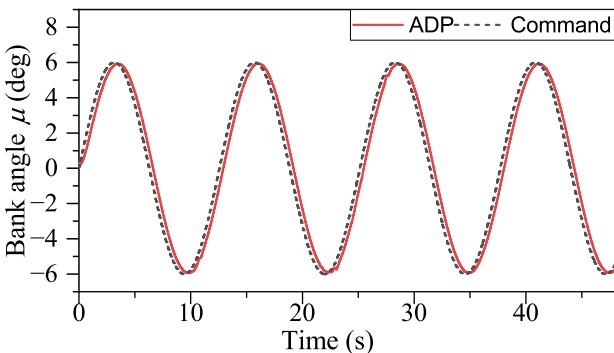

**Figure 20.** Bank angle.

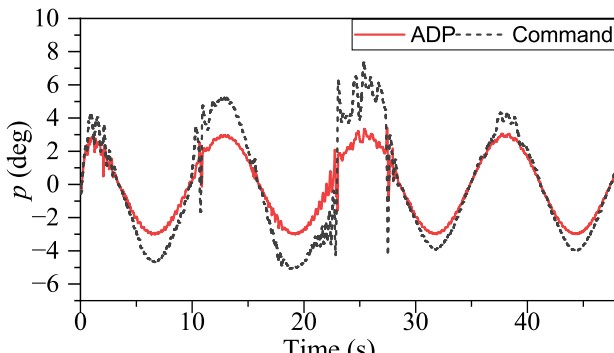

**Figure 21.** Body-axis roll rate.

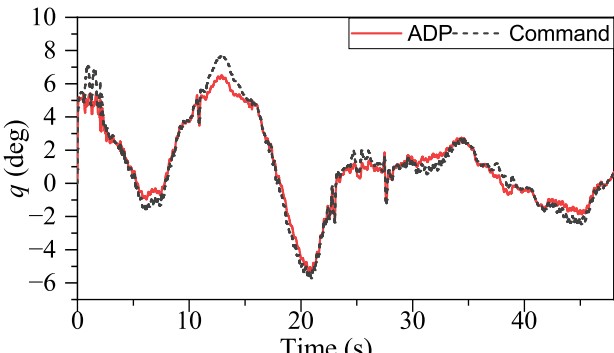

**Figure 22.** Body-axis pitch rate.

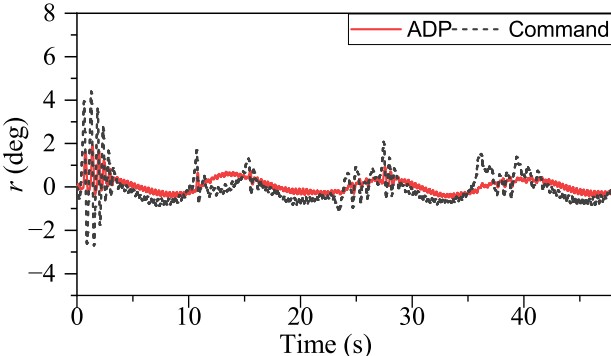

**Figure 23.** Body-axis yaw rate.

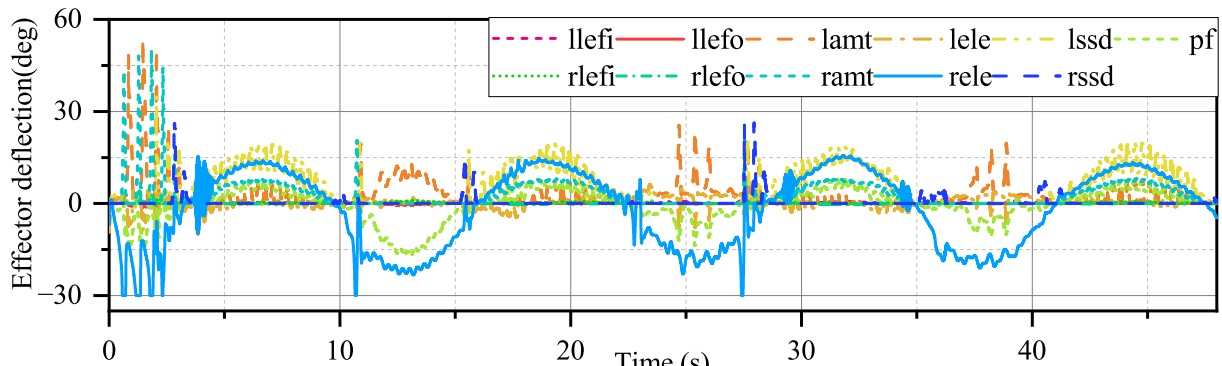

**Figure 24.** Effector deflection.

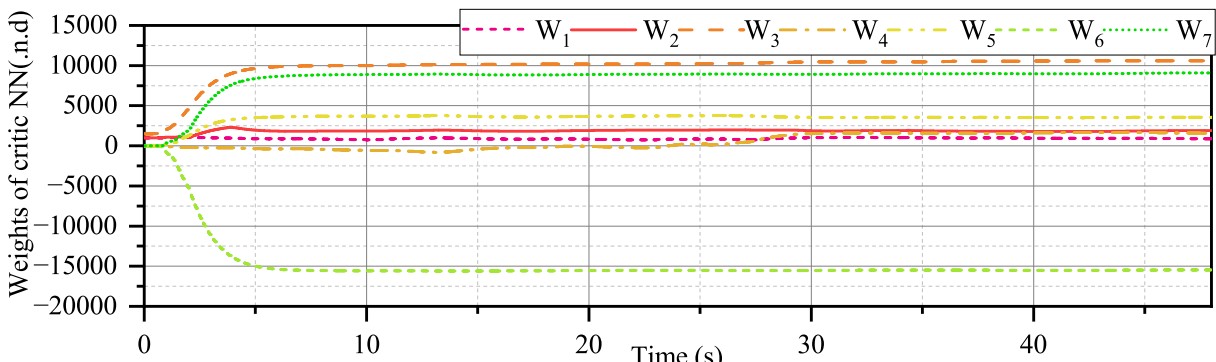

**Figure 25.** Convergence of the critical NN weights when $\bar{\delta} = [17.8, 17.8, \cdots, 17.8]^T$.

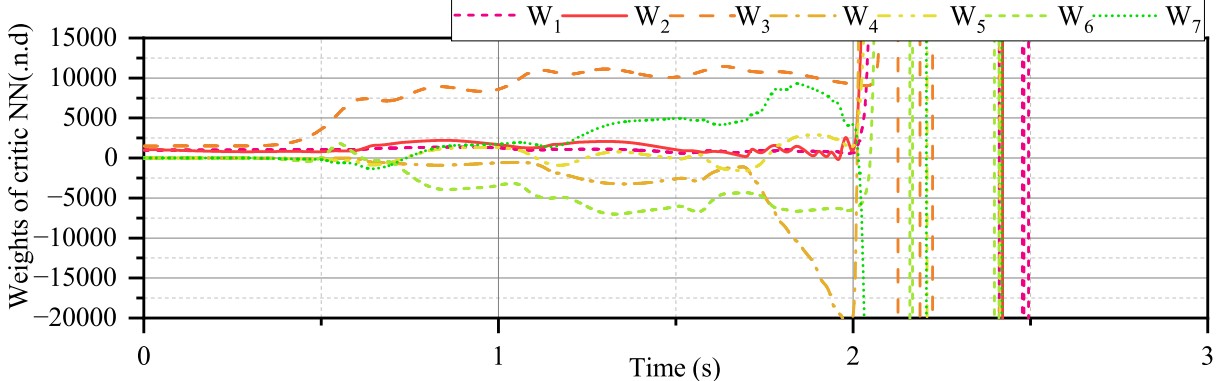

**Figure 26.** Convergence of the critical NN weights when $\bar{\delta} = [5.6, 5.6, \cdots, 5.6]^T$.

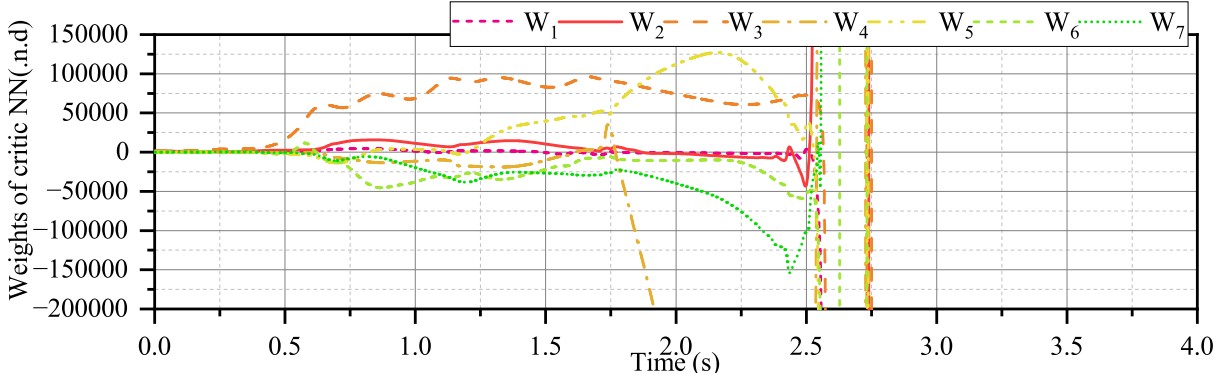

**Figure 27.** Convergence of the critical NN weights when $\bar{\delta} = [56.4, 56.4, \cdots, 56.4]^T$.

## 6. Conclusions and Outlook

The proposed method uses ADP to integrate control and control allocation, resulting in superior performance compared to conventional methods. Without using any model identification techniques, the ADP-based method exhibits strong convergence and robustness in the face of external disturbance and model uncertainty. Additionally, it presents a novel approach to flight control for over-actuated UAVs with nonaffine control inputs. From a control-theory perspective, the paper presents a straightforward yet efficient optimal tracking method for nonaffine systems, with theoretical evidence verifying its robustness. Specifically, this study has two key advantages in comparison to existing research. First, our method achieves better performance than traditional control architectures that separate control and control allocation by using a more optimized approach. Second, unlike many current optimal controllers for nonaffine systems, our method remains robust and does not depend on any model identifiers.

The proposed method has certain limitations that require attention. First, this method is only aimed at the cruise stage, as the nonlinear characteristics of the aircraft during this phase are not as prominent, and the optimal value function is relatively simple and can be well-fitted by a polynomial network. However, if large maneuvering flight is required, a more complex network structure needs to be introduced. This inevitably requires an improvement in the weight update rate to ensure system stability. Second, selecting the initial value for the critical network can be challenging when a complex network is used since the convergence of this method relies on the proper choice of the network's initial value. Thirdly, the design of the $\bar{\delta}$ is still heavily reliant on empirical knowledge. As demonstrated in the simulation section, a $\bar{\delta}$ that is too small may weaken robustness, while too large $\bar{\delta}$ may harm the closed-loop stability. Lastly, there is a dearth of real-world validation of this method. The external perturbations applied in the simulation offer only a limited exhibition of robustness and stability since the external interferences experienced by a UAV, in reality, are much more complex.

The next-step studies should focus on the following aspects: First, more complex neural networks can be introduced to further approximate the value function and handle more complex situations. Second, it would be very worthy work to introduce some intelligent algorithms to help design the performance function. Thirdly, only linear filters, as shown in Equation (34), are used in this paper, making our method better when facing high-frequency disturbances. In future studies, more advanced filters could be introduced to improve the performance of the ADP-based method when facing various disturbances. More importantly, it would be very expected that the performance of our method can be validated in real flight experiments.

**Author Contributions:** Conceptualization, Z.H. and Y.W.; Methodology, J.H.; Software, Z.H.; Validation, Y.B. and J.C.; Resources, J.H.; Data curation, Y.W.; Writing—original draft preparation, Z.H.; Writing—review and editing, Y.B. and L.H.; Visualization, Z.H.; Supervision, J.H.; Project administration, Z.H.; Funding acquisition, Y.W. All authors have read and agreed to the published version of the manuscript.

**Funding:** This work was supported by the National Natural Science Foundation of China [grant number 62103439], China Postdoctoral Science Foundation [grant number 2020M683716], and the Natural Science Basic Research Program of Shaanxi Province [grant number 2021JQ-364].

**Data Availability Statement:** Not applicable.

**Conflicts of Interest:** The authors declare no conflict of interest.

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
