# Peer review of "Attitude-Tracking Control for Over-Actuated Tailless UAVs at Cruise Using Adaptive Dynamic Programming"

_drones, doi:10.3390/drones7050294_

Round 1
Reviewer 1 Report
Dear Authors.
I revised the manuscript very carefully, and I think that is a valuable paper. The presented topic is not new, but you proposed a reasonable method. However, in the presented form, the manuscript cannot be accepted for publication. I suggest a major revision. Below please find my comments.
In the beginning, the state-of-the-art was presented. The contribution of the paper was clearly indicated. The ICE aircraft model was described. This is an over-actuated system, and the control is not trivial. The main issue of this paper is to solve the HJB equation and obtain the optimal value function. Finally, the two representative simulations were presented to illustrate the effectiveness of the proposed ADP-based integrated-control-and-control-allocation scheme. The paper ends with a summary of the main findings and suggestions for further possible research directions.
I did not detect plagiarism, and it seems that this is a novel and original paper. The English language requires moderate changes. I have found several minor issues:
Line 118
Sction 3 gives the problem… ----> Section 3 gives the problem…
Line 249
obtain by solving recatii equation. ----> obtain by solving Ricatii equation. (this is unclear)
There is a lot of similar issues.
The paper is written in a rigorous manner with step-by-step explanations of the methodology used.
General comments:
· The presented results should be reproducible by the other researchers. I suggest adding more details about the ICE aircraft. For example, some aerodynamic characteristics could be found in the following paper (you have also indicated this paper in the text):
Niestroy, M. A., Dorsett, K. M., & Markstein, K. (2017). A Tailless Fighter Aircraft Model for Control-Related Research and Development. AIAA Modeling and Simulation Technologies Conference. doi:10.2514/6.2017-1757
However, to perform the full 6-DoF numerical simulation the full set of the necessary data (mass, inertia matrix, aerodynamic chord, deflection ranges of the movable aerodynamic control surfaces, etc.) must be presented or at least mentioned where to find them. Please clearly indicate some papers/reports with the aircraft data in section “2. Model description”. In that way, it will be easier to find the necessary parameters.
· Could you provide more details about the model implementation? What language was used to implement the mathematical model? Which kind of solver was used (fixed/variable step solver)?
· In section “2. Model description” you presented the mathematical model of the aircraft. Could you provide a picture with the coordinate systems that were used in the modeling process? I think you used NED and body fixed-frames but it should be described.
· Could you provide a brief description of the main assumptions of the model?
· In chapter “5. Simulation verification” you presented some interesting results. However, the description and interpretation of these results are very messy. You should describe the results step-by-step in the appropriate order.
· The conclusion is very general and rather too short. It might be unclear how the proposed method overcomes other existing approaches.
Specific comments:
· The paper layout should be formatted according to the template delivered by MDPI. The manuscript requires extensive editing to make it publishable in the MDPI Drones. I highlighted some of the editing issues in the attached PDF file.
· In the “1. Introduction” you should cite the works using Author names. For example, instead of “[14] relaxed the need” it should be “He et al. [14] relaxed the need...”. Please check all these issues.
· In the figures, please change the hyphen (-) into a minus sign ($-$, "U+2212"). I think that the Editor might ask you about this issue in the proofreading process.
· The bibliography must be modified. MDPI citation style should be used. I suggest using a citation manager like Zotero or Mendeley.
· In this paper, you used a lot of symbols. I suggest introducing the list of symbols (together with units) in the attachment at the end of the paper.
Please consider my comments. I think that the paper might be accepted for publication when you will solve abovementioned issues.
Kind regards,
Reviewer
Author Response
Reply to the First Reviewer's Comments
Dear reviewer
Thank you for taking the time to review our manuscript and for providing us with such valuable feedback. We appreciate your insightful comments and suggestions, which will undoubtedly improve the quality of our paper. In response to your comments, we have made the following modifications to the paper, and corresponding changes are marked in red in the text:
For the General comments:
For the comment, 'The English language requires moderate changes. I have found several minor issues:'.
ans: We have carefully revised and polished the language of our paper and have made the necessary changes for the language errors, such as the errors you pointed out on lines 118 and 249.
For the comment, 'The presented results should be reproducible by the other researchers. I suggest adding more details about the ICE aircraft. For example, some aerodynamic characteristics could be found in the following paper (you have also indicated this paper in the text):'
ans: We present the basic parameters of ICE aircraft in Table 1, and in the first paragraph of Section 1, we added this sentence 'More detailed model information of the ICE aircraft, such as the modeling of effectors, etc., can be found in Chapter of [Stolk, A. Minimum drag control allocation for the Innovative Control Effector aircraft 2017.]. For reasons of space, this information is not repeated here.'
For the comment, 'Could you provide more details about the model implementation? What language was used to implement the mathematical model? Which kind of solver was used (fixed/variable step solver)?'
ans: We construct the aircraft model using Simulink and Matlab. The solver is Fix-step ode4(Runge-Kutta), and the Fixed-step size is 0.01s, we have added this in the first paragraph of Section 5, 'The simulations are conducted using Simulink, with the fixed-step ode4(Runge-Kutta) solve. The fixed-step size is 0.01 s.' Such descriptions have been added in line 402.
For the comment, 'In section “2. Model description” you presented the mathematical model of the aircraft. Could you provide a picture of the coordinate systems that were used in the modeling process? I think you used NED and body fixed-frames but it should be described.'
ans: The Equ.1 is defined in tangent-plane coordinate system, which is aligned as a geographic system but has its origin fixed at a point of interest on the spheroid; the Equ. 2 is defined in wind-axes system and Equ. 5 is defined in body-fixed coordinate system. The relationship between the wind-axes system and the body-fixed coordinate system is shown in Figure 1. The origin of both is at the aircraft's center of gravity, but the X-axis of the body-fixed coordinate system points in the direction of the nose, and the X-axis of the wind-axes system points in the direction of the relative wind. We have added this paragraph around line 210.
For the comment, 'Could you provide a brief description of the main assumptions of the model?'
ans: The modeling of the vehicle is based on the following two assumptions: first, the aircraft flies in the atmosphere, and the atmosphere is incompressible; second, the aircraft's body is rigid. Note that only the body of the aircraft is considered a rigid body, but the effectors are deformable. Such a description is added around line 190.
For the comment, 'In chapter “5. Simulation verification” you presented some interesting results. However, the description and interpretation of these results are very messy. You should describe the results step-by-step in the appropriate order'.
ans: We have restructured the language of the simulation section.
Specifically, the simulation 1 aims to illustrate the advantages of the proposed method compared to the traditional INDIPI. To facilitate the reader's understanding of the advantage of the proposed method, we have included the following language near line 444 as an outline for this subsection, 'The proposed method outperforms INDIPI in three ways: superior flight quality, intelligence, and better effectors deflection pattern.'
The simulation 2 is to test the robustness of the proposed method. Since the robustness of the proposed methods comes from \bar{\delta} , we have completely rearranged this subsection, and the content can be divided into three clearly structured layers, the first describing the experimental results, the second analyzing the effects of too small \bar{\delta} values, and the third analyzing the effects of too large \bar{\delta} values. I hope this could help readers understand our idea better.
For the comment, 'The conclusion is very general and rather too short. It might be unclear how the proposed method overcomes other existing approaches.'
ans: We have already rearranged Section. The following description was also included to illustrate the superiority of this study compared to existing studies.' Specifically, this study has two key advantages in comparison to existing research. Firstly, our proposed method achieves better performance than traditional control architectures that separate control and control allocation by utilizing a more optimized approach. Secondly, unlike many current optimal controllers for nonaffine systems, our method does not depend on any model identifiers and remains robust.'
For the specific comments:
For the comment, 'The paper layout should be formatted according to the template delivered by MDPI. The manuscript requires extensive editing to make it publishable in the MDPI Drones. I highlighted some of the editing issues in the attached PDF file.'
ans: We have already reformatted the paper according to the MDPI template.
For the comment, ' In the “1. Introduction” you should cite the works using Author names. For example, instead of “[14] relaxed the need” it should be “He et al. [14] relaxed the need...”. Please check all these issues.'
ans: We have cited most of the works in the introduction sections using Author names. Corresponding changes are marked in red in the text. But to maintain narrative fluency, when the citation is to highlight the techniques involved in the text, we still omitted the author's name.
For the comment, 'In the figures, please change the hyphen (-) into a minus sign (-, "U+2212"). I think that the Editor might ask you about this issue in the proofreading process.'
ans: Thanks for your kind reminder. We only detect the issue of hyphens and minus in Figure 1, and we have already changed the hyphen into minus
For the comment, 'The bibliography must be modified. MDPI citation style should be used. I suggest using a citation manager like Zotero or Mendeley.'
ans: The bibliography has already been modified into MDPI citation style.
For the comment, ' In this paper, you used a lot of symbols. I suggest introducing the list of symbols (together with units) in the attachment at the end of the paper.'
ans: Thank you for the reminder. We have added the Nomenclature in Table 1 and Table 2 around Line 194. We hope this could help with reading.

Reviewer 2 Report
The paper proposes an attitude tracking scheme that integrates control and control allocation for the over-actuated tailless aircraft featured by nonlinearity and nonaffine control inputs.
I have a problem with the review of this work. I appreciate the effort of the authors in the field of control theory, but they definitely do not see the need for validation of the method on real drones.
The authors write: "In terms of practical use, the proposed method provides a new idea for the flight control of over-actuated aircraft with nonaffine control inputs", and then specify in Conculsions three ideas for the development of the method itself, and do not even mention the need robust validation of what they did in flight conditions.
Simple simulation models (transfer function with band-limited white noise pass the second order filter) are not convincing.It is difficult to draw such far-reaching conclusions based on only two sample simulations.
I did not find typos. The article is edited legibly and correctly. I think the selection of literature is accurate.
Why is it not on the MDPI template?
Author Response
Dear Reviewer,
Thank you for reviewing our paper and providing valuable feedback. We appreciate your concern about the validation of our proposed method on real drones. We agree that flight testing is important for the validation of control methods, but at this stage, it is not feasible for us to conduct such experiments due to resource limitations. However, we want to clarify that the dynamic model used in our simulations is based on real wind tunnel data, which ensures the reliability of our simulation experiments.
Regarding the robustness verification of our method, we acknowledge that our trigonometric-function external disturbance was not comprehensive enough to cover all possible scenarios. However, there are many studies that have adopted similar external disturbances to validate the robustness of control methods, which are shown below. We do recognize that theoretical robustness verification in simulation has limitations, but it is an important step in the development of control methods before flight testing.
[1] HOU Y, LV M, LIANG X, 2022. Fuzzy adaptive fixed-time fault-tolerant attitude tracking control for tailless flying wing aircrafts[J/OL]. Aerospace Science and Technology, 130: 107950. https://doi.org/10.1016/j.ast.2022.107950. DOI:10.1016/j.ast.2022.107950.
[2]WANG Y, WU Q, 2017. Adaptive non-affine control for the short-period model of a generic hypersonic flight vehicle[J/OL]. Aerospace Science and Technology, 66: 193–202. http://dx.doi.org/10.1016/j.ast.2017.03.005. DOI:10.1016/j.ast.2017.03.005.
[3]WANG Y, HU J, ZHENG Y, 2019. Improved decentralized prescribed performance control for non-affine large-scale systems with uncertain actuator nonlinearity[J/OL]. Journal of the Franklin Institute, 356(13): 7091–7111. https://doi.org/10.1016/j.jfranklin.2019.03.032. DOI:10.1016/j.jfranklin.2019.03.032.
[4]WANG Y, HU J, WANG J, 2018. Adaptive neural novel prescribed performance control for non-affine pure-feedback systems with input saturation[J/OL]. Nonlinear Dynamics, 93(3): 1241–1259. https://doi.org/10.1007/s11071-018-4256-4. DOI:10.1007/s11071-018-4256-4.
[5]WANG Y, HU J, 2018. Improved prescribed performance control for air-breathing hypersonic vehicles with unknown deadzone input nonlinearity[J/OL]. ISA Transactions, 79(May): 95–107. https://doi.org/10.1016/j.isatra.2018.05.008. DOI:10.1016/j.isatra.2018.05.008.
[6]WANG Y, HU J, LI J, 2019. Improved prescribed performance control for nonaffine pure-feedback systems with input saturation[J]. International Journal of Robust and Nonlinear Control, 29(6): 1769–1788. DOI:10.1002/rnc.4466.
[7]HE Z, HU J, WANG Y, 2022. Sample entropy based prescribed performance control for tailless aircraft[J/OL]. ISA Transactions, (xxxx). https://doi.org/10.1016/j.isatra.2022.04.041. DOI:10.1016/j.isatra.2022.04.041.
[8]HE Z, HU J, WANG Y, 2022. Incremental Backstepping Sliding-Mode Trajectory Control for Tailless Aircraft with Stability Enhancer[J]. Aerospace, 9(7). DOI:10.3390/aerospace9070352.
[9]BU X, LV M, LEI H, 2023. Performance for Waverider Vehicles : A Fragility-avoidance Approach[J]. (March). . DOI:10.1109/TCYB.2023.3255925.
[10]BU X, LV M, LEI H, 2023. Performance for Waverider Vehicles : A Fragility-avoidance Approach[J]. (March). . DOI:10.1109/TCYB.2023.3255925.
As for your comment on "transfer function with band-limited white noise pass the second order filter," we actually use this method to generate command signals to demonstrate that the proposed method can track command with unknown dynamic, highlighting its superiority over most of existing optimal trackers that can only track commands with known dynamic. It is not intended to verify the robustness of the proposed method.
We appreciate your positive comments regarding the quality of our writing and literature selection. Regarding the MDPI template, we apologize for not adhering to it and will make the necessary changes before resubmitting the paper.
More specifically, we have made the following changes to the paper,
In conclusion section, line 562, 'More importantly, it would be very expected that the performance of the proposed method can be validated in real flight experiment.'
Once again, thank you for your time and constructive feedback.
Sincerely,

Reviewer 3 Report
1. This study presents an attitude tracking control for over-actuated tailless aircraft at cruise using adaptive dynamic programming. The proposed method was verified through two simulations and compared to traditional control architectures. However, the readability and organization of this article need to be improved since there are too many repeat sentences.
2. The contribution and novelty of this article are not clear. It is suggested to rewrite the abstract and conclusion parts.
3. The saturations of the control surfaces (effectors) in angular rates are required since the fluctuating in Figure 14 around fifth sec is unrealistic for real aircraft. It is suggested to add the saturations of the angular rates for all control effectors in simulation environment.
4. It is suggested to discuss the limitations and weaknesses of the proposed method, especially for the parameter tuning and robustness in different dynamic motions.
5. Some representations of the variables must be checked again such as Line 154.
Author Response
Reply to the Third Reviewer's Comments
Dear reviewer,
Thank you very much for providing the review comments on our article. We would like to express our sincere appreciation for the constructive feedback that has enabled us to improve the quality of our manuscript.
In response to your comments, we have made the following changes to the paper:
For the comment, '1. This study presents an attitude tracking control for over-actuated tailless aircraft at cruise using adaptive dynamic programming. The proposed method was verified through two simulations and compared to traditional control architectures. However, the readability and organization of this article need to be improved since there are too many repeat sentences.'
ans: We have thoroughly polished our submission, with particular emphasis on refining the abstract, summary, and Introduction section. We hope this could help readers.
For the comment, 'The contribution and novelty of this article are not clear. It is suggested to rewrite the abstract and conclusion parts.'
ans: We have already rewritten the abstract and conclusion parts, as shown as below:
Abstract
Using adaptive dynamic programming, this paper presents a novel attitude tracking scheme for over-actuated tailless unmanned aerial vehicles (UAVs) that integrates control and control allocation while accounting for nonlinearity and nonaffine control inputs. The proposed method utilizes the idea of nonlinear dynamic inversion to create an augmented system and converts the optimal tracking problem into an optimal regulation problem using a discounted performance function. Drawing inspiration from incremental control, the proposed method achieves optimal tracking control for the nonaffine system by simply using a critic-only structure. Besides, the unique design of the performance function ensures robustness against model uncertainties and external disturbances. The proposed method was found to outperform traditional control architectures that separate control and control allocation, achieving the same level of attitude tracking performance through a more optimized approach. Furthermore, unlike many recent optimal controllers for nonaffine systems, the proposed method does not require any model identifiers and demonstrates robustness. The superiority of the proposed approach is verified through two simulated scenarios, and the internal mechanism of the proposed method is further discussed. The theoretical analysis of the robustness and stability of the proposed method is also provided.
Conclusion and outlook
The proposed method utilizes ADP to integrate control and control allocation, resulting in superior performance compared to conventional methods. Without using any model identification techniques, the proposed method exhibits strong convergence and robustness in the face of external disturbance and model uncertainty. Additionally, it presents a novel approach to flight control for over-actuated UAV with nonaffine control inputs. From a control theory perspective, the paper presents a straightforward yet efficient optimal tracking method for nonaffine systems, with theoretical evidence verifying its robustness. Specifically, this study has two key advantages in comparison to existing research. Firstly, our proposed method achieves better performance than traditional control architectures that separate control and control allocation by utilizing a more optimized approach. Secondly, unlike many current optimal controllers for nonaffine systems, our method remains robust and does not depend on any model identifiers.
The proposed method has certain limitations that require attention. Firstly, the proposed method is only aimed at the cruise stage, as the nonlinear characteristics of the aircraft during this phase are not as prominent and the optimal value function is relatively simple and can be well-fitted by a polynomial network. However, if large maneuvering flight is required, a more complex network structure needs to be introduced. This inevitably requires an improvement in the weight update rate to ensure system stability. Secondly, selecting the initial value for the critic network can be challenging when a complex network is used since the convergence of the proposed method relies on the proper choice of the network's initial value. Thirdly, the design of the \bar{\boldsymbol{\delta}} is still heavily reliant on empirical knowledge. As demonstrated in the simulation section, a \bar{\boldsymbol{\delta}} that is too small may weaken robustness, while too large \bar{\boldsymbol{\delta}} may harm the closed-loop stability. Lastly, there is a dearth of real-world validation of the proposed method. The external perturbations applied in the simulation offer only a limited exhibition of the proposed method's robustness and stability since the external interferences experienced by a UAV, in reality, are much more complex.
The next-step studies should focus on the following aspects: Firstly, more complex neural networks can be introduced to further approximate the value function and handle more complex situations. Secondly, designing a suitable performance function is not easy, especially for the nonaffine system covered in this paper. It would be very worthy work to introduce some intelligent algorithms to help design the performance function. Thirdly, only linear filters, as shown in Eq.3, are used in this paper, making the proposed method better when facing high-frequency disturbances. In future studies, more advanced filters could be introduced to improve the performance of the proposed method when facing various disturbances. More importantly, it would be very expected that the performance of the proposed method can be validated in real flight experiment.
For the comment, 'The saturations of the control surfaces (effectors) in angular rates are required since the fluctuating in Figure 14 around the fifth sec is unrealistic for real aircraft. It is suggested to add the saturations of the angular rates for all control effectors in the simulation environment.'
ans:
We apologize for any confusion or inconvenience caused by Figure 14. In previous versions, we did not mention the amplitude and rate limits of the effectors because our proposed method was only facing the cruising stage, and the method itself can reduce control inputs, so the effectors were not likely to be saturated. The unrealistic appearance of Figure 14 was mainly due to the width of the lines. In our revised manuscript, we have added a description of the capabilities of the effectors and their implementation methods in the simulation, as in line 193:
The deflection ranges of the effectors are, inboard LEF: 0-40 deg, outboard LEF: \pm40 deg, ELE: \pm30 deg, PF: \pm30 deg, AMT: \pm60 deg, SSD: 0-60 deg. The rate limits on the leading edge devices are 40 deg/sec and all the other surfaces as 150 deg/sec.
and line 417:
It is assumed that the leading edge actuators are represented by the transfer function \frac{(18)(100)}{((s+18)(s+100))} while all the other actuators, including thrust vectoring, as \frac{(40)(100)}{((s+40)(s+100))}.
and we have redone all the simulations. For any figures involving control inputs, such as Figures 15, 16, and 24, we have replaced the lines with thinner ones to help readers better understand our ideas. Once again, we apologize for any confusion and appreciate your constructive feedback.
For the comment, 'It is suggested to discuss the limitations and weaknesses of the proposed method, especially for the parameter tuning and robustness in different dynamic motions.'
ans: We have added the discussion of the limitations and weaknesses of the proposed method, as shown as in Line 556
"The proposed method has certain limitations that require attention. Firstly, the proposed method is only aimed at the cruise stage, as the nonlinear characteristics of the aircraft during this phase are not as prominent and the optimal value function is relatively simple and can be well-fitted by a polynomial network. However, if large maneuvering flight is required, a more complex network structure needs to be introduced. This inevitably requires an improvement in the weight update rate to ensure system stability. Secondly, selecting the initial value for the critic network can be challenging when a complex network is used since the convergence of the proposed method relies on the proper choice of the network's initial value. Thirdly, the design of the \bar{\boldsymbol{\delta}} is still heavily reliant on empirical knowledge. As demonstrated in the simulation section, a \bar{\boldsymbol{\delta}} that is too small may weaken robustness, while too large \bar{\boldsymbol{\delta}} may harm the closed-loop stability. Lastly, there is a dearth of real-world validation of the proposed method. The external perturbations applied in the simulation offer only a limited exhibition of the proposed method's robustness and stability since the external interferences experienced by a UAV, in reality, are much more complex."

Round 2
Reviewer 2 Report
Unfortunately, I do not consider the authors' comments to be accurate. The general remark: "it is not feasible for us to conduct such experiments due to resource limitations" without giving specific reasons, confirms my belief that this is a paper maybe more for a conference? or to a control theory journal? instead of the MDPI Drones, where I, as a reviewer and reader of papers, expect at least a "proof of concept", validated on real drones. Even the best simulation experiment do not take into account such factors as the stability of communication, constant data transmission rate and calculations in the real-time manner, the changes of the thrust force value depending on the level of energy in the power supply system, etc., many of engineering problems where one can only see whether a given method has implementation potential or not.
Unfortunately, there is still a large gap between the PID-type controllers commonly used in UAVs and sophisticated theoretical and simulation work, which can only be filled by validation on real drones. And this is needed for the development of mobile robotics. Therefore, I recommend taking your time with this work, implementing the results in hardware-in-the-loop conditions, and then making test flights compared to standard approaches.
Author Response
Dear reviewer
We greatly appreciate your constructive and forward-thinking suggestions. We agree that hardware-in-the-loop experiments in real-world conditions are essential for validating our proposed controller. We also acknowledge the importance of addressing engineering problems like communication stability, data transmission rates, and power supply dependencies in real flight tests. Applying the proposed method to actual UAVs is exactly what we are currently working hard to do. The proposed method in this paper, although currently only validated through simulation, represents a significant milestone in our research plan and a critical cornerstone for advancing toward practical flight applications.
In the recent study of ICE, high-performance simulation verification is widely accepted by most researchers, both in theoretical research and engineering practice. For journals that are more geared towards engineering practice, such as MDPI Aerospace, Aerospace Science Technology, and Chinese Journal of Aeronautics, there have been published research results using simulation validation similar to our submission, for example:
[1]He, Z., Hu, J(2022). Incremental Backstepping Sliding-Mode Trajectory Control for Tailless Aircraft with Stability Enhancer. Aerospace.
[2]Cong, J., Hu(2023). Fault-Tolerant Attitude Control Incorporating Reconfiguration Control Allocation for Supersonic Tailless Aircraft. Aerospace.
[3]Hou, Y., Lv, M., Liang, X., & Yang, A. (2022). Fuzzy Adaptive Fixed-Time Fault-Tolerant Attitude Tracking Control for Tailless Flying Wing Aircrafts. Aerospace Science and Technology.
[4]Sun, L., Zhou, Q., Jia, B., Tan, W., & Li, H. (2020). Effective control allocation using hierarchical multi-objective optimization for multi-phase flight. Chinese Journal of Aeronautics, 33, 2002-2013.
Specifically, in our submission, we have compared our method with the above Cong's method and achieved better performance. Additionally, many theses in the aerospace engineering field from Delft University of Technology also use simulation validation, such as:
[5]VAN OORSPRONK D J, 2018. Upset Recovery Controller Based on Maximum Control Effectiveness for a High-Performance Over-Actuated Aircraft[J]. : 106. .
[6]BAKKER J M, 2019. Benchmark flight scenarios for testing fault tolerant control in high performance aircraft[J]. Master’s Thesis, Delft University of Technology, Delft, the Netherlands.
[7]STOLK A R J, 2017. Minimum drag control allocation for the Innovative Control Effector aircraft[J]. : 142. .
[8]AARSSEN M S T van den, 2020. Distributed Approach for Aerodynamic Model Identification of the ICE Aircraft[J].
[9]MATAMOROS I, 2017. Nonlinear control allocation for a high-performance tailless aircraft with innovative control effectors - An incremental robust approach[J]. Master’s Thesis, Delft University of Technology, Delft, the Netherlands, : 107.
[10]SHAYAN K. MSc Thesis Report Application of Continuous Reinforcement Learning on Innovative Control Effector Aircraft MSc Thesis Report Application of Continuous Reinforcement Learning on Innovative Control Effector Aircraft[J].
In comparison to previous studies, we present a novel control architecture that integrates control and control allocation. We apply ADP for its solution and achieve better performance at a lower cost. Therefore, we believe that our research not only has rich theoretical value but will also be recognized by researchers in the field of engineering practice, inspiring drone engineers to develop more creative control mechanisms and serving as a catalyst for further development.
Besides, our paper is to be submitted to Drones Journal's special issue 'Flight control system simulation', so we believe the selection of the journal is correct.
Thank you again for your valuable feedback.
The author
Author Response
Dear Reviewer,
Thank you very much for your careful review of our paper. We have carefully checked our submission again and have corrected errors similar to bar\delta. We also have rewritten the conclusion section of Simulation 2. We believe that the revised words is more organized and will be easier for readers to understand the main point of our paper. The modifications can be found in lines A to B, and the revised content is as follows:
The above result shows that \boldsymbol{\bar{\delta}} must exceed the upper limit of external disturbance effects so that the proposed method can show robustness.
However, \boldsymbol{\bar{\delta}} should not exceed a reasonable value either. To illustrate this point, we conducted a convergence test of the critic neural network with \boldsymbol{\bar{\delta}} = [56.4,56.4,\cdots,56.4]^T, which is very large. The result is depicted in Fig.27. It can be observed that the system experiences a significant shift in critic NN weights, leading to a collapse within 3 seconds.
From the analysis of Eq.34, having a too large \boldsymbol{\bar{\delta}} can result in \mathbf{P} not being positive-definite. More bluntly, too large \boldsymbol{\bar{\delta}} could dominate the dynamics of critic neural network weights, causing the ignored of sampled data that aid in policy improvement.}
In conclusion, Simulation 2 demonstrates that the proposed method can effectively withstand model uncertainties and external disturbances, given the appropriate selection of \boldsymbol{\bar{\delta}}.
Thank you again for your helpful feedback, which has enabled us to improve the quality of our paper.
Sincerely,
The author
